# Context Similarity Structure Shapes the Emergence of Reliable In-Context and In-Weights Mixtures

## Abstract

We aim to train models that co-develop in-context learning (ICL) and in-weights learning (IWL), and flexibly switch between them based on context relevance. Such models should exploit closely related in-context examples while relying on IWL when examples are irrelevant. Although LLMs exhibit both modes, standard task-specific fine-tuning often erodes ICL, motivating IC-Train, a form of fine-tuning with in-context examples. When trained under IC-Train, prior work has shown that emergence of ICL depends on factors such as task diversity and training duration. We show that an overlooked factor is the similarity structure between target inputs and context examples. Of the two existing modes of context-target pairing, random context leads to IWL dominance, while only similar examples in context causes ICL to degenerate to copying labels without regard to relevance. To address this, we propose Contrastive-Context which enforces two types of contrasts: (1) mix of similar and random examples within a context to evolve a correct form of ICL, and (2) varying grades of similarity across contexts to evolve IWL-ICL mixtures. With experiments on real sequence to sequence learning tasks on four models, we show that Contrastive-Context strengthens ICL while preserving IWL, outperforming random and nearest-neighbor sampling in both in-domain and out-of-domain evaluation. Theoretical analysis and diagnostic probes confirm that contrasted contexts yield stable ICL–IWL mixtures, avoiding collapse into pure ICL, IWL, or copying. Our results establish similarity structure as a key driver of reliable ICL under fine-tuning an LLM for a task.

## 1 Introduction

Our goal is to train models on tasks with limited labeled data so they can continuously incorporate feedback in the form of new examples at test time, without further training. Such adaptation should boost accuracy on inputs resembling the new examples, while retaining generalization to inputs without close neighbors. In-context learning (ICL) offers a natural mechanism for flexibly absorbing labeled data by placing test inputs alongside labeled examples. Pre-trained LLMs Brown et al. (2020) show strong ICL ability, yet prior studies Alves et al. (2023); Duan et al. (2024); Wang et al. (2024); Goyal et al. (2025), as well as our experiments, find that fine-tuning often diminishes it. A natural alternative is to fine-tune in ICL mode (IC-Train), but this is challenging due to competition with in-weights learning (IWL). Moreover, ICL emergence is transient, shaped by factors like training steps and task diversity Chan et al. (2024); Singh et al. (2025); Nguyen & Reddy (2025); Wurgaft et al. (2025); Ku et al. (2025). Most studies explore synthetic tasks where diversity can be controlled. In contrast, we study a stable real task, where ICL is needed not to learn a new task but to flexibly and continually absorb examples. Here, models must operate as ICL–IWL mixtures—adapting through ICL when context is relevant, relying on IWL for examples without close neighbors, and deciding at inference which to use.

In this paper, we show that a critical but overlooked factor in training viable ICL-IWL mixtures is the similarity structure between the target and in-context examples. Two standard defaults, viz, random sampling and selecting the $k$ nearest examples, push models to opposite extremes:

1. With random examples as context, IWL dominates and prior ICL ability is lost.

2. With overly similar examples, IWL is diminished and the model learns to copy labels, ignoring input relevance, and engaging in a degenerate form of ICL.

To address this, we propose a training strategy Contrastive-Context that samples examples for the contexts so as to (i) span multiple similarity levels to the target and (ii) contrast in similarity among themselves. When highly similar real examples are unavailable to create the contrast, we generate synthetic in-context examples via small perturbations of the target.

We evaluate Contrastive-Context on machine translation (MT) between English and other languages at various grades of exposure to the LLM. Recently Ramos et al. (2025) have shown that for MT, instruction tuning is less effective than learning from examples. Across four models (1B–8B), we show that Contrastive-Context consistently strengthens ICL while preserving IWL, outperforming both random and nearest-neighbor sampling under in-domain and out-of-domain evaluation. On the entire spectrum of target-context relatedness, Contrastive-Context provides gains over standard zero-shot fine-tuning, whereas other forms of IC-Train perform worse than zero-shot in at least one end of the context-target similarity spectrum.

We theoretically analyze a minimal two-layer transformer, and show that a stationary point of the self-attention layer is an optimal mixture of ICL and IWL when trained with contrasted contexts. To extend the understanding to LLMs, we design probes to disentangle ICL, IWL, and blind-copying, and observe the emergence of ICL-IWL mixtures when fine-tuning LLMs with Contrastive-Context, while settling for the corner points of ICL, IWL, or blind-copying for other context regimes. Our study establishes inter-example and example-target similarity as a key driver of whether fine-tuning enhances, erodes, or deforms ICL capabilities and mixes with in-weights learning.

**Our contributions.** (1) We identify inter-example similarity as a critical but underexplored factor shaping the emergence (or erosion) of ICL during fine-tuning with in-context examples. (2) We propose Contrastive-Context, a training strategy that samples examples across similarity levels within and across contexts and injects synthetic perturbations when needed. (3) We empirically evaluate the methods on four 1B–8B models over four machine translation tasks, eleven Text-to-SQL task, three multilingual semantic parsing tasks, and two synthetic alignment reasoning tasks. Our experiments show that Contrastive-Context consistently improves accuracy across diverse in-context configurations and domains. (4) We theoretically analyze the three context regimes on a minimal two-layer transformer to provide insights on the critical role of both inter and intra context contrasts for evolving ICL-IWL mixtures. (5) We empirically study emergence of different forms of learning on real models and tasks with three well-designed probes, showing how Contrastive-Context enables ICL–IWL mixtures without collapsing into one of pure IWL, pure ICL, or blind copying.

## 2 RELATED WORK

Starting from the landmark study of Brown et al. (2020) showing the emergence of ICL in pre-trained LLMs, the topic of In-context learning has been extensively studied along various aspects including understanding ICL emergence Garg et al. (2022); Zhang et al. (2023a); Olsson et al. (2022); Shi et al. (2024); Agarwal & Sarawagi (2025), evaluating ICL on real tasks Kossen et al. (2024); Bertsch et al. (2024), and instance selection during deployment Levy et al. (2023); Kothyari et al. (2025). Our focus is on understanding the interplay between ICL and IWL during task-specific training with in-context examples. In this mode, each training instance $(\mathbf{x}_*, \mathbf{y}_*)$ is preceded with $k$ related input-output pairs $(\mathbf{x}_1, \mathbf{y}_1), (\mathbf{x}_2, \mathbf{y}_2), \ldots, (\mathbf{x}_k, \mathbf{y}_k)$. Prior work has studied how ICL-IWL emergence is influenced by two kinds of relatedness of the target to in-context examples.

**Relatedness of the x to y mapping function.** Many prior work study a setup where training is over a mixture of $T$ tasks sampled from a task family (say linear regression with task-specific weights). For the $k$ examples within a training instance, the output $\mathbf{y}_i$ is determined by the same task $f_\tau(\mathbf{x})$. During IC-Train with a task mixture, a model is said to develop ICL if it uses the in-context examples to infer the $f_\tau$, and IWL if the $T$ tasks are learned in parameters of the LLM. Several studies show that as task diversity ($T$) increases, models prefer ICL over IWL Reddy (2024); Singh et al. (2024); Akyürek et al. (2024); Edelman et al. (2024); Park et al. (2025); Nguyen & Reddy (2025); Wurgaft et al. (2025); Ku et al. (2025); Kim et al. (2025). The tasks considered are synthetic regression, classification, or sequence completion Park et al. (2025); Akyürek et al. (2024); Rajaraman et al. (2024); Edelman et al. (2024). Nguyen & Reddy (2025) show that IWL and ICL rely on different parameters with distinct learning rates, making ICL emergence task-dependent.

Singh et al. (2025) instead argue for a cooperative scenario where ICL transitions to a context-conditional IWL. All these works use random selection of in-context examples from $P(X)$. Fu et al. (2024) show that mixing diverse tasks from a random pool, with a fixed task promotes both ICL and IWL, which they demonstrate on scalar labeling tasks. Our work departs by studying IC-Train on a single real sequence to sequence task and isolating how inter-example relatedness shapes the ICL–IWL tradeoff.

**Relatedness of x tokens.** Chan et al. (2022) show that when the data distribution is bursty, causing related examples to appear in the context, the model develops ICL capabilities, whereas for non-bursy distributions IWL emerges — ICL and IWL develop together when the data follows a Zipfian distribution. Singh et al. (2023) further observe that ICL is a transient phenomenon, and asymptotically could reduce to IWL. Zucchet et al. (2025) discuss how token repetition in context speeds up the emergence of ICL. Bratulić et al. (2025) show that repetition of Query-Label pairs in-context promotes ICL. All these studies are on scalar classification or regression tasks. Our work focuses on sequence prediction tasks, and we present a finer-grained analysis in terms of similarity that generalizes repetition. Further, we stress the importance of contrast within the in-context examples to promote ICL over blind copying. In machine translation, Alves et al. (2023) show that fine-tuning LLMs in ICL mode preserves ICL ability, often lost during standard zero-shot fine-tuning. We show that poorly chosen context examples can harm ICL even more than zero-shot.

Overall, our work presents a finer-grained analysis of the impact of context-target and inter-example similarity in the emergence of ICL-IWL mixtures when fine-tuning a model on a specific task.

## 3 Impact of Training Strategies on Generalization Across Target-Context Relatedness

Let $P_\theta$ denote a model to be trained on a task (e.g. low-resource translation). Typically, $P_\theta$ will be a pre-trained LLM. We are given a labeled dataset $D$ of $N$ pairs of inputs-outputs $\{(\mathbf{x}_1, \mathbf{y}_1), \ldots, (\mathbf{x}_N, \mathbf{y}_N)\}$ drawn from the underlying joint distribution $P(X, Y)$ of this task. As is often the case with today's large models, $P_\theta$ may have been partially exposed to $P(X, Y)$ during pre-training. Our goal is to train $P_\theta$ using $D$ so that its performance remains robust under these deployment settings: (1) As more labeled pairs are appended to $D$ (e.g., via user feedback), the model's accuracy should improve on test inputs with highly similar cases in $D$, *without requiring further parameter updates.* (2) For test inputs lacking similar examples in $D$, accuracy should be no worse than that of a model trained in the standard zero-shot setting. As discussed earlier, a natural candidate to meet these goals is IC-Train. In standard fine-tuning, we would minimize loss on each example independently as $\max_\theta \mathbb{E}_{(\mathbf{x},\mathbf{y})\sim D} \log P_\theta(\mathbf{y}|\mathbf{x})$. By contrast, in IC-Train, we sample $k+1$ labeled examples from $D$, place the first $k$ pairs as in-context examples while imposing loss on the target:

$$\text{IC-Train:} \quad \max_\theta \mathbb{E}_{\{(\mathbf{x}_i, \mathbf{y}_i): i=1\ldots k+1\}\sim D} \log P_\theta(\mathbf{y}_{k+1}|[\mathbf{x}_1, \mathbf{y}_1, \ldots \mathbf{x}_k, \mathbf{y}_k, \mathbf{x}_{k+1}]) \tag{1}$$

The above training explicitly trains the model to leverage in-context examples, thereby better preparing it to absorb labeled data that may arrive during deployment. This paper shows that a crucial but often overlooked factor in IC-Train is the inter-example similarity among the $k+1$ samples. To make this explicit, we rewrite the IC-Train objective in terms of the strategy $\texttt{Choose}(D, \mathbf{x})$, which determines how the $k$ examples are chosen from $D$ to accompany a target input $\mathbf{x}_*$:

$$\max_\theta \mathbb{E}_{(\mathbf{x}_*, \mathbf{y}_*)\sim D} \mathbb{E}_{\{(\mathbf{x}_i, \mathbf{y}_i): i=1\ldots k\}\sim \texttt{Choose}(D, \mathbf{x}_*)} \log P_\theta(\mathbf{y}_*|[\mathbf{x}_1, \mathbf{y}_1, \ldots \mathbf{x}_k, \mathbf{y}_k, \mathbf{x}_*]) \tag{2}$$

As we will show, $\texttt{Choose}(\mathbf{x}_*, D)$ critically influences whether IC-Train strengthens or erodes our desired robustness properties.

Most prior work has studied IC-Train in the setting where the $k$ examples in a context $C$ are chosen at random independent of $\mathbf{x}_*$. We refer to this method as **Random-Context**. A second strategy is to select as $C$ the top-$k$ examples from $D$ that are most similar to $\mathbf{x}_*$, which we call **Similar-Context**. As we show in Section 3.1, both these strategies fail to meet the robustness goals outlined earlier, albeit for different reasons. Under Random-Context, the model relies heavily on in-weights learning and fails to benefit from new related examples that appear in the context. Under Similar-Context training, the model learns to exploit labels in context without adequately judging whether the corresponding input $\mathbf{x}_i$s are relevant to the $\mathbf{x}_*$. This causes accuracy to suffer on test examples

without close neighbors, and in extreme cases can cause the model to just blindly copy the $\mathbf{y}_i$ labels from the context. We therefore propose an alternative strategy, which we call Contrastive-Context.

**Contrastive-Context:** The key idea is to pair target instances with contexts so as to create contrast both within examples in a context and across contexts. We achieve this in two steps: First, we select a **single** example, say $(\mathbf{x}_i, \mathbf{y}_i)$ for an arbitrary $i \in [1, k]$ at one of three similarity levels: (1) Most-similar: with probability $p > 0$ choose the most similar example from $D$, (2) Highly-similar: with small probability $\epsilon$, create a synthetic highly similar example by small perturbation of $\mathbf{x}_*$. This last step is needed when the training data $D$ is small, and many examples fail to find close neighbors in the second step. (3) Random: with a probability $1 - p - \epsilon$ choose a random example from $D$. For NLP tasks, the perturbation is obtained by getting a paraphrase of $\mathbf{x}_*$. We use $p = 0.5, \epsilon = 0.1$ in all experiments in the paper[1]. In the second step, we place this selected example within a contrastive set of unrelated examples by sampling the remaining $k - 1$ examples randomly from $D$. We later show that a similar example juxtaposed with unrelated ones within the same context forces the model to harness in-context labels only after establishing similarity to the target. And varying similarity levels across batches forces the model to balance in-weights learning with in-context learning. We provide theoretical insights on the need for both intra and inter context contrast in Section 4.1.

### 3.1 EMPIRICAL COMPARISON OF DIFFERENT TRAINING STRATEGIES

We empirically compare IC-Train under the three types of context and standard Zero-Context training on their accuracy across varied contexts ranging from random to highly relevant. We fine-tune four open source models on four real sequence-to-sequence learning tasks, and evaluate under both an in-domain and out-of-domain test-set for a total of 32 (model,task,test-set) configurations.

**Datasets.** We translate from English to four different languages: Lithuanian, Tamil, Hindi, and German that cover a wide spectrum of an LLM's exposure to these languages. Table 1 in the Appendix provides more information. For testing, we consider two types: an In-Distribution (ID) dataset, and to evaluate the capability of continuous adaptation, an out-of-distribution dataset (OOD) from specialized domain such as judiciary and religion. In each case, in-context examples are chosen from a held-out set (e.g., Flores dev split for Flores devtest). To obtain paraphrases for Contrastive-Context, we employed the `gemini-2.0-flash-lite` model. For Similar-Context, we used the `all-MiniLM-L6-v2` model to further filter the training instances, retaining only those with an average similarity of the few-shot examples with $\mathbf{x}_*$ greater than $0.5$. For all training setups (Random-Context, Similar-Context, Contrastive-Context) we choose $k$ randomly from $[0 \ldots 5]$.

**Evaluation Setup.** In order to study the effect of different grades of relatedness of the target test input to the in-context examples, each test example in $D$ is evaluated under three different contexts: (1) Randomly selected $k$ examples from $D$, (2) Select $k$ examples most similar to $\mathbf{x}_*$ using BM25, (3) $k - 1$ random examples and one closely related example obtained to be a paraphrase of $\mathbf{x}_*$ with $\mathbf{y}_i = \mathbf{y}_*$. This mode checks the scenario of whether a model can harness closely related feedback from users after training. Over the union of all (context,test) instances, we plot accuracy as a function of the maximum similarity of the input $\mathbf{x}_*$ to an example in its context. We use COMET-22 Rei et al. (2020) to measure translation accuracy.

**Models.** We experiment with the following open-source LLMs: 1. Llama 3.2 1B base, 2. Llama 3.2 8B Instruct Grattafiori et al. (2024) 3. Qwen 2.5 7B Yang et al. (2024). 4. Mistral 7B v0.3 Jiang et al. (2023) The exact prompt used for obtaining the translations in ICL mode is given in Appendix Section A. For training, we adopted LoRA with a rank of 16, a scaling factor of $\alpha = 32$, and a dropout rate of $0.05$. Training was performed with a batch size of 2 using Adam with a learning rate of $2 \times 10^{-4}$ and a linear decay learning rate scheduler.

In Figure 1 we plot accuracy against the maximum target-context similarity (found using the `all-MiniLM-L6-v2`[2] model) on combinations of four models, four tasks, two test settings, and five training methods: Zero-Context, IC-Train with Random-Context, Similar-Context, Contrastive-Context, and the original untuned base model. For clarity we group similarity values into three bins

---

[1]In Appendix Section D.1 we show ablation with other choices.
[2]https://huggingface.co/sentence-transformers/all-MiniLM-L6-v2

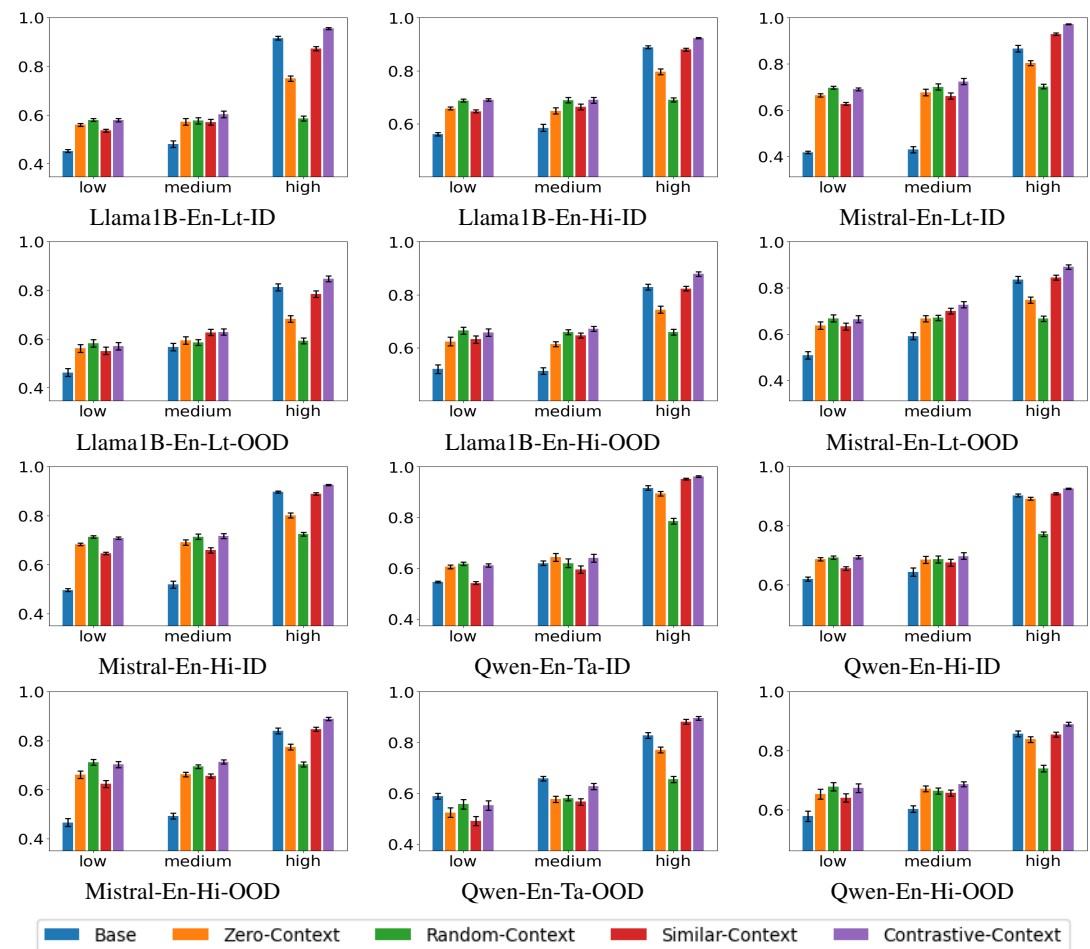

Figure 1: Effect of fine-tuning a base model with different strategies (Zero-Context, and IC-Train under Random-Context, Similar-Context, and Contrastive-Context) on accuracy over varying grades of similarity to in-context examples for 32 different models, language-pairs, and ID/OOD datasets. Remaining plots in Appendix Figure G. X-axis: Level of maximum similarity of target with in-context examples. The similarity ranges here are - Low: $0-0.33$, Medium:$0.33-0.67$, High: $0.67-1$. Y-axis: Accuracy (COMET score). Main observations: **Contrastive-Context competitive or more accurate across the entire spectrum of target-context relatedness. On targets with high context similarity, model fine-tuned with Zero-Context is worse than baseline, Random-Context even worse than Zero-Context. On targets with low context similarity, Similar-Context is worse than Zero-Context and Random-Context.**

of Low, Medium, and High similarity. The error bars shows the 95% confidence interval with 1000 bootstrap resampling of the test data. Raw plots and over all 32 task-model combinations appear in Appendix Figure G. Based on these plots, we make the following observations:

1. Zero-Context (**orange** bar) boosts accuracy of the untuned base model (**blue** bar) for low to medium target-context similarity. But it causes loss of ICL ability of the base model as seen by the drop in accuracy for high target-context similarity cases (right side of each plot) in almost all 32 model-task-test settings. This is in agreement with the conclusions of earlier studies Alves et al. (2023); Duan et al. (2024); Wang et al. (2024).

2. IC-Train with Random-Context (**green** bar) provides the least gains in accuracy with increased target-context similarity. This holds across all models (big and small) in both ID and OOD settings. Its performance is worse than even Zero-Context when presented with similar examples in context! This shows that Random-Context cannot support OOD generalization or harness highly related examples in-context.

3. IC-Train with Similar-Context (**red** bar) suffers in the low similarity region, and is worse than all other forms of fine-tuning in that region. It provides decent accuracy in medium similarity range, but its capability to harness highly related examples in context is worse than baseline's for almost all model language combinations in both ID and OOD settings.

4. Contrastive-Context (**violet** bar), is among the highest performing methods in all test settings in both ID and OOD deployment, and across all models. Compared to Similar-Context, the second best performer, Contrastive-Context scores when highly related examples are present in context, while being competitive in low similarity ranges.

## 3.2 EXPERIMENTS ON OTHER TASKS

We present experiments on three other task types: Text-to-SQL, Multilingual Semantic Parsing, and an Alignment Reasoning Task (in Appendix C.3).

**Text to SQL.** We experiment on Text-to-SQL as an instance of a Text-to-code generation task where the need for online adaptation to private databases is compelling. We use the BIRD dataset Li et al. (2024b) for Text2SQL, with its official train split for fine-tuning and dev split spanning eleven distinct databases. More details in Appendix C.2. The results shown in Figure 2(a) further add evidence to the robustness of Contrastive-Context to handling contexts at varying similarity levels. Observe how IC-Train with Random-Context is worse than even Zero-Context in the high similarity range, and with Contrastive-Context we obtain the best accuracy across all levels. In this task Similar-Context does not suffer in the low-similarity range because the labeled pool is small, and Similar-Context almost reduces to random for many instances.

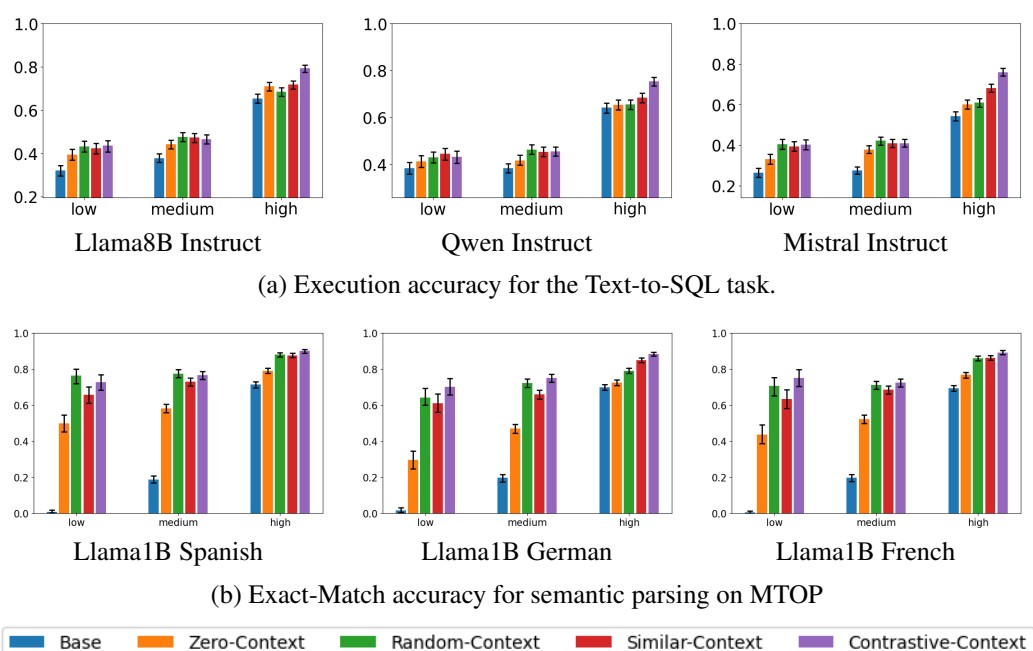

(a) Execution accuracy for the Text-to-SQL task.

(b) Exact-Match accuracy for semantic parsing on MTOP

Figure 2: Accuracy ($Y$-axis) against three levels of similarity ($X$-axis). The similarity ranges here are - Low: $0 - 33$, Medium: $33 - 67$, High: $67 - 100$. Error bars show 95% confidence intervals.

**Multilingual Semantic Parsing.** We use the MTOP dataset from XSemPLR Zhang et al. (2023b), a unified benchmark for cross-lingual semantic parsing. We experiment with three languages — Spanish, German, and French. The results are shown in Figure 2(b). Observe that Similar-Context suffers in the low and medium similarity regions compared to Random-Context and Contrastive-Context. On the other had, in the high similarity region, Random-Context suffers compared to Similar-Context and Contrastive-Context. In this task too Contrastive-Context performs competitively or better across all similarity levels.

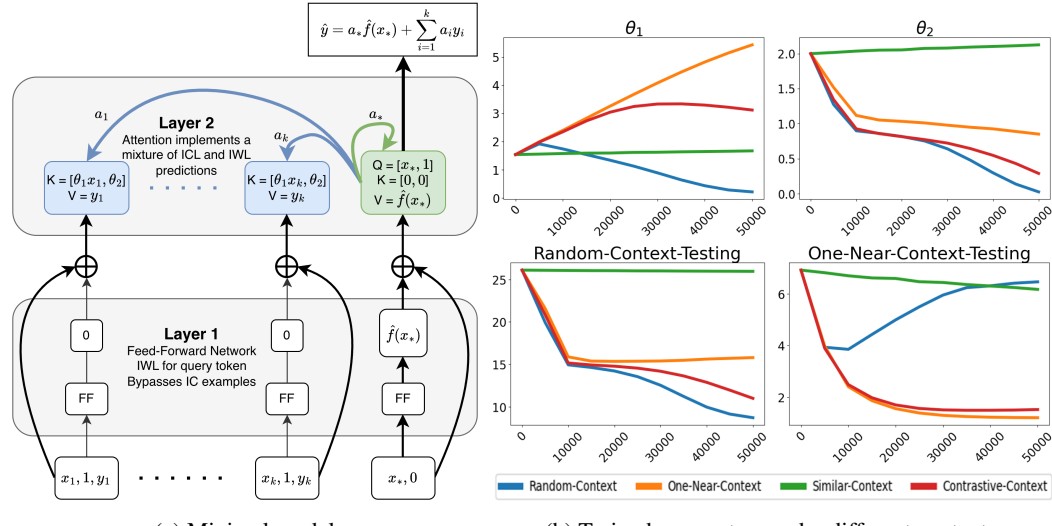

(a) Minimal model                    (b) Trained parameters under different contexts.

Figure 3: (a) Schematic of a minimal two-layer transformer with an in-weights learner ($\hat{f}$) in layer-1 and a two parameter second layer that implements the ICL-IWL mixtures. The parameter $\theta_1$ drives correct type of ICL while $\theta_2$ drives blind copying. (b) X-axis: Training steps. Y-axis: Value of $\theta_1, \theta_2$ (in the upper two graphs) or MSE loss (in the lower two graphs) for different training regimes. Observe how in Random-Context $\theta_1, \theta_2$ are both decreasing, in Similar-Context they are both increasing, in One-near and Contrastive-Context they move in opposite directions as is ideal. This explains the high loss of Similar-Context and One-near-Context when tested with random context and of Random-Context and Similar-Context when tested with One-Near context. Only Contrastive-Context provides low loss in both settings.

## 4 EXPLAINING DIFFERENCES IN IC-TRAIN UNDER DIFFERENT CONTEXTS

In this section we seek to explain why the three different forms of training lead to such difference in their generalization across different levels of relatedness of in-context examples. Specifically, we will show that IC-Train with Random-Context reduces to IWL, with Similar-Context reduces to ICL or blind-copying, and with Contrastive-Context learns to switch between ICL and IWL depending on context-target similarity. In order to understand such behavior conceptually we design a minimal model of a two-layer transformer and theoretically analyze its stationary points under various training regimes on a simple regression task. Later in Section 4.2 we support the insights gained via empirical evidence on real translation tasks over pre-trained LLMs.

### 4.1 THEORETICAL ANALYSIS WITH A MINIMAL MODEL

Let input covariates $\mathbf{x} \in \mathbf{R}^d$ be sampled from a distribution $P(X)$ where $\|\mathbf{x}\|_2 = 1$, with labels given by $y = f(\mathbf{x})$, where $f(\mathbf{x})$ denotes the function to be learned. Our goal is to learn $f$ using a Transformer trained using IC-Train under the three context sampling regimes.

Figure 3 provides an overview. The transformer takes $k + 1$ inputs: first $k$ examples $\{(\mathbf{x}_i, y_i)\}_{i=1}^{k}$ form the context $C$, and then the target input $\mathbf{x}_*$. We design a minimal two-layer Transformer that performs two steps. We defer detailed realization of the parameters of such a transformer in Appendix E. First, a feed-forward block of the lower layer computes an in-weights estimate $\hat{f}(\mathbf{x}_*)$ from the query token's initial features. The final attention layer performs a selective computation at the query position governed by two learnable scalar parameters, $\theta_1$ and $\theta_2$, which control the trade-off between ICL and IWL. The steps in this layer are:

1. Compute attention scores for all context tokens ($a_1, \ldots, a_k$). The unscaled attention score $s_i$ for each token $i$ is determined by the similarity $\theta_1 \mathbf{x}_*^\top \mathbf{x}^i$ plus a global bias $\theta_2$. Here, $\theta_1$ acts as an inverse temperature, controlling how sharply the model focuses on the most similar context examples. The value vector for position $i$ is the token's label, $y_i$.

2. Compute a *self-attention* score ($a_*$) from the query to itself, with a fixed logit of 0. The bias $\theta_2$ controls the overall tendency to use the context. A more negative $\theta_2$ suppresses all context scores, favoring the self-path. The value vector for the last position is the in-weights prediction, $\hat{f}(\mathbf{x}_*)$.

3. Compute the output of the attention layer at the query position as the attention weighted sum of these value vectors. The prediction for a target $\mathbf{x}_*$ is thus:

$$\hat{\mathbf{y}} \;=\; a_* \hat{f}(\mathbf{x}_*) + \sum_{i=1}^{k} a_i y_i, \;\; \text{where } s_i = \exp\left(\theta_1 \mathbf{x}_i^\top \mathbf{x}_* + \theta_2\right), \;\; S = \sum_{i=1}^{k} s_i, \;\; a_i = \frac{s_i}{1+S}, \;\; a_* = \frac{1}{1+S}$$

We analyze the stationary points of the squared loss under different context selection strategies.

$$\mathcal{L}(\theta_1, \theta_2, \hat{f}) = \mathbb{E}_{\mathbf{x}_* \sim D, C \sim \texttt{Choose}(D, \mathbf{x}_*)} \left[ (f(\mathbf{x}_*) - \hat{\mathbf{y}})^2 \right]$$

**Assumptions for Analysis**  Our analysis relies on the following assumptions.

- **[Lipschitz]** The ground-truth function $f$ is $L$-Lipschitz: for all $\mathbf{x}, \mathbf{x}'$, $|f(\mathbf{x}) - f(\mathbf{x}')| \leq L\|\mathbf{x} - \mathbf{x}'\|_2$.
- **[Context Regimes]** Fix small parameters $0 \leq \delta_1, \delta_2 \ll 1$. For any target $\mathbf{x}_*$, the context selection procedure $\texttt{Choose}(D, \mathbf{x}_*)$ yields one of the following regimes:
  (i) *Random-Context:* $\forall i \in [1, k]$, $\mathbf{x}_i^\top \mathbf{x}_* \leq \delta_1$.
  (ii) *Similar-Context:* $\forall i$, $\mathbf{x}_i^\top \mathbf{x}_* \geq 1 - \delta_2$.
  (iii) *One-Near-Context:* For one $j^\star \in [1, k]$ $\mathbf{x}_{j^\star}^\top \mathbf{x}_* \geq 1 - \delta_2$ and for all $i \neq j^\star$, $\mathbf{x}_i^\top \mathbf{x}_* \leq \delta_1$.
  (iv) *Contrastive-Context:* With probability $p$, a training instance has a Random-Context, and with probability $1 - p$, it has a One-Near-Context.
- **[In-weights MSE Comparison]** Let $E = \mathbb{E}_D[(\hat{f}(\mathbf{X}) - f(\mathbf{X}))^2]$ be the population MSE of the in-weights estimator. We assume that due to a limited training budget, this estimator is outperformed by ICL from a very similar example but is better than ICL from a random example. That is, $E$ is bounded by $L^2\|\mathbf{x}_i - \mathbf{x}_*\|_2^2$. For a similar point, $\|\mathbf{x}_i - \mathbf{x}_*\|_2^2 = 2(1 - \mathbf{x}_i^\top \mathbf{x}_*) \leq 2\delta_2$. For a random point, $\|\mathbf{x}_i - \mathbf{x}_*\|_2^2 \geq 2(1 - \delta_1)$. Overall, we get these dataset-dependent bounds:

$$2L^2 \delta_2 \;\leq\; E \;\leq\; 2L^2(1 - \delta_1) \tag{A}$$

**Optimal Parameters for Different Regimes**  We now analyze the stationary points[3] of the loss $\mathcal{L}$ for each context regime. Detailed proofs of stationarity and optimality are in the Appendix.

**Case 1: Random-Context.** When all context examples are dissimilar to the target, the optimal strategy is to ignore the context and rely on in-weights learning.

- **Optimal Parameters:** The limit $\theta_2 \to -\infty$ is a stationary point. This forces all attention scores $s_i \to 0$, causing the attention weight on the in-weights prediction to dominate ($a_* \to 1$).
- **Resulting Loss:** The prediction becomes $\hat{\mathbf{y}} \to \hat{f}(\mathbf{x}_*)$, and the loss is $\mathcal{L}_{\text{param}} = E$.
- **Brittleness:** With a *One-near* context, it would fail to use the highly relevant example achieving a sub-optimal loss of $E$ instead of the ICL loss $\leq 2L^2\delta_2$, which is lower as per Eqn A.

**Case 2: Similar-Context.** When all context examples are highly similar to the target, the model should perform ICL by averaging the context labels.

- **Optimal Parameters:** The limit $\theta_1 + \theta_2 \to \infty$ is a stationary point. This drives all scores $s_i \to \infty$ at a comparable rate, causing the weight on the in-weights prediction to vanish ($a_* \to 0$).
- **Resulting Loss:** The prediction becomes a weighted average of context labels, $\hat{\mathbf{y}} \to \sum_i a_i y_i$, with a low loss of $\mathcal{L}_{\text{icl}} \leq 2L^2\delta_2$.
- **Brittleness:** This model may learn to always trust the context, if it reaches $\theta_1 + \theta_2 \to \infty$ using $\theta_2 \to \infty$. When given a *Random-Context*, it would still average the random labels: $a_* = 0, a_i = \frac{1}{k} \forall i \implies \hat{y} = \sum_i y_i / k$. This leads to a high error, close to $2L^2(1 - \delta_1)$ worse than IWL (Eqn A).

**Case 3: One-Near-Context.** Here, the optimal strategy is to copy its label.

- **Optimal Parameters:** The limit $\theta_1 \to \infty$ while $\theta_1 + \theta_2 \to \infty$ is a stationary point. This makes the score $s_{j^\star}$ for the near point dominate all others, so that $a_{j^\star} \to 1$.

---

[3]Note that while our analysis uses the theoretical limits of $\theta_i \to \pm\infty$, these represent optimization directions. In practice, due to the exponential scaling in the softmax, the desired behavior of attention weights saturating at 0 or 1 is achieved once the parameters $\theta_i$ attain a sufficiently large finite magnitude.

- **Resulting Loss:** The prediction converges to $\hat{\mathbf{y}} \to y_{j^\star}$, with loss $\mathcal{L}_{\text{icl}} \le 2L^2\delta_2$.
- **Brittleness:** This model learns a "copy-the-best" heuristic. In a *Random-Context*, the large $\theta_1$ amplifies small differences in $\mathbf{x}_i^\top \mathbf{x}_*$, causing it to copy the closest label, which is still random.

**Case 4: Contrastive-Context.** Here the model should learn an ICL-IWL mixture that can switch between the two based on the context.

- **Optimal Parameters:** The stationary point that emerges is the limit where $\theta_1 \to \infty$ and $\theta_2 \to -\infty$ such that: (a) $\theta_1(1-\delta_2) + \theta_2 \to \infty$ (for the near point) (b) $\theta_1\delta_1 + \theta_2 \to -\infty$ (for the random points). This can be achieved by setting $\theta_2 = -c\theta_1$ for a constant $c$ where $\delta_1 < c < 1 - \delta_2$.
- **Adaptive Behavior:** This parameter setting produces optimal behavior in all regimes, overcoming the brittleness of specialized models. • Under *Random-Context*, condition (b) forces $a_* \to 1$, correctly defaulting to the in-weights prediction $\hat{\mathbf{y}} \to \hat{f}(\mathbf{x}_*)$. • Under *One-Similar* context, conditions (a) and (b) force $a_{j^\star} \to 1$, correctly switching to the ICL prediction $\hat{\mathbf{y}} \to y_{j^\star}$. • Under *Similar-Context*, it behaves like the model trained on *Similar-Context* due to condition (a), thus averaging all the labels.

The above theoretical analysis reveals the importance of creating contrast both within examples in a context and across contexts to ensure that the model learns to harness context only based on similarity to the target, and to rely on IWL when context examples are not similar enough.

### 4.2 Empirical Study of Emergence of Different Forms of Learning on LLMs

In the above minimal model, just two parameters $\theta_1, \theta_2$ were enough to understand the effect of target-context similarity on the emergence of different forms of learning. To extend this understanding to large multi-billion parameter models on real sequence-to-sequence tasks, we design three different external probes, and analyze these scores as training progresses.

#### PROBES TO DETECT DIFFERENT FORMS OF LEARNING

Let $\hat{\mathbf{y}}_C = \text{argmax}_{\mathbf{y}} P_\theta(\mathbf{y}|C, \mathbf{x}_*)$ denote the predicted output under set $C = [\mathbf{x}_1, \mathbf{y}_1, \dots \mathbf{x}_k, \mathbf{y}_k]$ as in-context examples and $\hat{\mathbf{y}}_\phi$ denote zero-shot prediction. Let $\mathcal{K}(\mathbf{x}, \mathbf{x}')$ be a measure of similarity between two inputs. We used cosine similarity of their sentence embeddings.

**In-Weights Learning Probe.** We quantify the in-weights learning as the similarity (measured with COMET) between predictions under random and empty context. IWL-score$(P_\theta) = \mathbb{E}_{\mathbf{x}_*} \mathbb{E}_{C\sim\text{Random}(D,k)} \text{sim}(\hat{\mathbf{y}}_C, \hat{\mathbf{y}}_\phi)$

**In-Context Learning Probe.** We quantify ICL capability by comparing prediction $\hat{\mathbf{y}}_C$ under a one-similar context $C$ to the labels in context weighted by their similarity to $\mathbf{x}_*$. For NLP tasks we obtain a one-similar context by injecting a paraphrase for $\mathbf{x}_*$ with gold $\mathbf{y}_*$ among remaining $k-1$ random examples, so $C = [\mathbf{x}_1, \mathbf{y}_1, \dots, \mathbf{x}_{i^*} = \text{paraphrase}(\mathbf{x}_*), \mathbf{y}_{i^*} = \mathbf{y}_*, \dots \mathbf{x}_k, \mathbf{y}_k]$ Here we use the COMET score as our similarity metric and the function $\mathcal{K}$ calculates the similarity between $x_*$ and $x_j$ ICL-score$(P_\theta) = \mathbb{E}_{\mathbf{x}_*} \mathbb{E}_{C\sim\text{One-similar}(D,\mathbf{x}_*,k)} \sum_{j=1}^{k} \frac{\mathcal{K}(\mathbf{x}_*,\mathbf{x}_j)\text{sim}(\hat{\mathbf{y}}_C,\mathbf{y}_j)}{\sum_i \mathcal{K}(\mathbf{x}_*,\mathbf{x}_i)}$

**Blind Copy Probe.** A model that develops the propensity to copy indiscriminately from the context would output a $\hat{\mathbf{y}}$ that is similar to a label $\mathbf{y}_i$ in-context irrespective of $\mathbf{x}_*$'s similarity to $\mathbf{x}_i$. We quantify this by shuffling the one-similar context (above) so that no $\mathbf{x}_i$ and $\mathbf{y}_i$ are correctly matched. An example for $k = 3$ is: $C = [\mathbf{x}_2, \mathbf{y}_*, \mathbf{x}_2 = \text{paraphrase}(\mathbf{x}_*), \mathbf{y}_3, \mathbf{x}_3, \mathbf{y}_1]$ A pure ICL model would output $\mathbf{y}_3$ based on similarity of $\mathbf{x}_2$ with $\mathbf{x}_*$ whereas a model that copies independent of $\mathbf{x}$ could output one of the other labels. To capture such behavior, we define copy score as maximum similarity to a label other than the one attached to $\mathbf{x}_{i^*}$ where $i^*$ is the position of $\mathbf{x}_*$'s paraphrase. Here we use the BLEU score as our similarity metric. Copy-score$(P_\theta) = \mathbb{E}_{\mathbf{x}_*} \mathbb{E}_{C\sim\text{Shuffle(One-similar}(D,\mathbf{x}_*,k))} \max_{i\in C,i\neq i^*} \text{sim}(\hat{\mathbf{y}}_C,\mathbf{y}_i)$

In Figure 4 we show the emergence of different forms of learning as measured by these three probes for various model, language pair, dataset combinations from Section 3.1. More combinations appear in the Appendix Figure H. We can make the following observations from these graphs: 1. The IWL-score (first column) of Random-Context, and Contrastive-Context steadily increases with training steps with Contrastive-Context lagging only slightly behind. However, for Similar-Context the IWL is distinctly lower, sometimes by a significant margin. This explains why Similar-Context provides lower accuracy in Figure 1 when target-context similarity is low. 2. The ICL-score (second

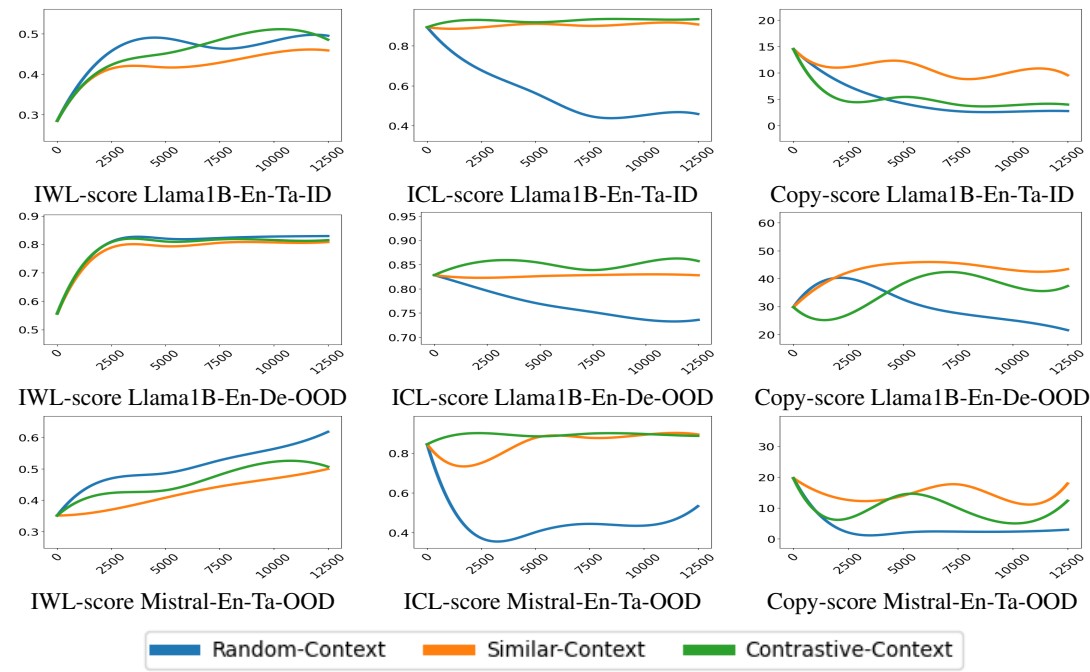

Figure 4: Emergence of different forms of learning in three different training methods: Random-Context, Similar-Context, and Contrastive-Context. X-axis is training steps and Y-axis denotes scores of one of the three probes. Results of other model-task and datasets in Appendix Figure H. IWL-score of Similar-Context is lowest, ICL-score of Random-Context diminishes fast with training, Copy-score of Similar-Context is higher. Contrastive-Context provides best retention of ICL and IWL capabilities without resorting to copying. The current plots have been smoothed out for better visualization.The unsmoothed plots are in Appendix Figure H.1

column) shows that for Random-Context ICL capability steadily drops with training, explaining why it provides almost the same accuracy across different grades of target-context similarity. For Contrastive-Context and Similar-Context, the ICL capability stays almost the same or increases. 3. The copy score (last column) decreases with training using Random-Context, and Contrastive-Context and gets substituted by either in-weights on in-context learning. In contrast, for Similar-Context the copy score is higher than for the other models. On real datasets, Similar-Context training may not consistently show increased copy tendencies because not every target will find all similar top-k examples, and the training data may naturally contain a mix of random and similar contexts. But in spite of the natural mixing, we observe Similar-Context to result in reduced IWL and more blind copying compared to Contrastive-Context.

## 5 CONCLUSION

We studied how to train models that balance in-context learning (ICL) with in-weights learning (IWL) and switch between them based on context relevance. Our analysis shows this balance is fragile, and strongly influenced by the target-context similarity patterns: random contexts lead to IWL dominance, while overly similar ones reduce ICL to degenerate copying. We introduced Contrastive-Context, a simple strategy of creating contrast both within examples in a context and across contexts. A theoretical analysis with a minimal transformer on regression tasks further highlighted why such contrasts are essential: they ensure models exploit context based on similarity of the inputs, while reverting to IWL when none of in-context examples are relevant. Experiments over 32 model–task–test settings for low-resource MT, a Text-2-SQL task, three multilingual semantic parsing tasks, and a synthetic alignment task demonstrate that contrasted contexts preserve IWL while sustaining robust ICL. Probes on large LLMs further confirm similarity structure as a decisive factor in avoiding collapse into pure IWL, pure ICL, or blind copying.

## REPRODUCIBILITY STATEMENT

All datasets are public and URLs to the data sources are present in Appendix Section C. Experimental details are present in Section 3.1. The scripts for our experiments will be released as soon as we get a chance to anonymize them and after the discussion forum opens. The prompts used with LLMs are present in Appendix Sections A and B. For the theoretical proofs, all assumptions are mentioned in Section 4.1, and detailed proofs are present in Appendix Section F.

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

## A PROMPTS FOR OBTAINING TRANSLATIONS WITH IN-CONTEXT EXAMPLES

```
• Translate the source text from English to German.
  Source: Redistribution is only possible if there is actually
  ↪  something produced to be redistributed.
  Target:
• Consider the following 4 translations from English to
  ↪  German.
  Example 1:
```

```
        Source: This is now to be given concrete shape in the
        ↪   proposal to give EU citizens the right of free movement
        ↪   and residence.
        Target: Jetzt soll sie in dem Vorschlag über das Recht der
        ↪   Unionsbürger auf Freizügigkeit und Aufenthalt konkret
        ↪   ausgestaltet werden.

        Example 2:
        Source: These data can only be seen as striking, alerting us
        ↪   to the urgent need for humanitarian aid to these
        ↪   countries to be properly directed to the area of health
        ↪   and the provision of basic medical care to highly
        ↪   deprived communities.
        Target: Diese Zahlen sind erschreckend und müssen für uns
        ↪   ein Alarmsignal sein, dass es dringend notwendig ist,
        ↪   die humanitäre Hilfe für diese Länder in angemessener
        ↪   Weise in den Bereich des Gesundheitswesens und der
        ↪   Bereitstellung von medizinischer Grundversorgung für die
        ↪   am meisten benachteiligten Gemeinschaften zu lenken.

        Example 3:
        Source: There are a number of actions we need to take in
        ↪   future in the area of budget control within the
        ↪   agencies.
        Target: Es gibt eine Reihe von Maßnahmen, die wir in Zukunft
        ↪   im Bereich der Haushaltskontrolle in den Agenturen
        ↪   ergreifen müssen.

        Example 4:
        Source: It is precisely in the area of the environment that
        ↪   people's dissatisfaction has been most deeply felt.
        Target: Gerade im Umweltbereich spürt man die
        ↪   Unzufriedenheit der Bevölkerung am deutlichsten.

        Translate the source text from English to German.
        Source: It is precisely in the area of budget policy that
        ↪   these three basic principles need to be given concrete
        ↪   expression.
        Target:
```

## B  PROMPT TO GENERATE PARAPHRASES

```
Generate 5 diverse paraphrases for the given input English
↪   sentence.
Before paraphrasing, carefully analyze the original sentences and
↪   keywords.
Ensure significant variation in phrasing, structure, and wording
↪   while preserving meaning.
The paraphrases should be such that their translation into another
↪   language could still be the same.
Only modify the grammar and sentence structure such that all the
↪   paraphrases should be translatable to a common sentence in
↪   another language.

### Inputs:
Original Sentences:
{sentences}
```

```
### Example:
1.
#### Input:
Original Sentences:
[
    "Sentence 1": "In New York City, the streets were alive with
    ↪  the sound of honking taxis and chatter from the crowded
    ↪  sidewalks.",
    "Sentence 2": "\"I can't believe it's already time for the
    ↪  annual reunion,\" Emily said, her voice filled with
    ↪  excitement.",
    "Sentence 3": "The Amazon rainforest is home to countless
    ↪  species of plants and animals, many of which have yet to
    ↪  be discovered.",
]

#### Example Paraphrases:
{{
    "Sentence 1": [
        "The streets of New York City buzzed with the noise of
        ↪  honking taxis and voices from the bustling
        ↪  sidewalks.",
        "In the heart of New York City, honking taxis and lively
        ↪  chatter filled the streets.",
        "New York City's streets were vibrant with the sounds of
        ↪  honking taxis and the chatter of crowded sidewalks.",
        "In the bustling streets of New York City, honking taxis
        ↪  and lively conversations created a lively
        ↪  atmosphere.",
        "The sound of honking taxis and the chatter from crowded
        ↪  sidewalks made the streets of New York City come
        ↪  alive."
    ],
    "Sentence 2": [
        "Emily exclaimed, her excitement evident, \"I can't
        ↪  believe it's time for the annual reunion already.\"",
        "With excitement in her voice, Emily said, \"I can't
        ↪  believe the annual reunion is here already.\"",
        "Emily's voice was filled with excitement as she said, \"I
        ↪  can't believe it's already time for the annual
        ↪  reunion.\"",
        "With a voice full of excitement, Emily remarked, \"I
        ↪  can't believe the annual reunion is already upon
        ↪  us.\"",
        "Filled with excitement, Emily exclaimed, \"I can't
        ↪  believe it's already time for the annual reunion!\""
    ],
    "Sentence 3": [
        "The Amazon rainforest shelters an untold number of plant
        ↪  and animal species, many still waiting to be
        ↪  discovered.",
        "Countless species of plants and animals call the Amazon
        ↪  rainforest home, many of which remain undiscovered.",
        "Home to countless species of plants and animals, the
        ↪  Amazon rainforest holds many that are yet to be
        ↪  discovered.",
        "The Amazon rainforest is a habitat for innumerable
        ↪  species of plants and animals, many of which are still
        ↪  unknown.",
```

```
          "Many species of plants and animals, yet to be discovered,
          ↪  thrive in the Amazon rainforest."
      ],
}}

### Output Instructions:
- Generate exactly 5 paraphrases.
- Ensure correct JSON output syntax like the example
- Always escape quotes within a paraphrase (like in example 2)
```

# C  DATASETS AND ADDITIONAL TASKS

## C.1  LOW RESOURCE TRANSLATION

Table 1: Datasets Used for MT. Flores+ refers to Flores-200. **Sources:** Europarl, Tanzil, and EMEA from Tiedemann (2012); Samanantar from Ramesh et al. (2022); Flores+ from NLLB Team et al. (2024); Judicial from Kunchukuttan et al. (2018).

| | Train | | ID Test | | OOD Test | |
|---|---|---|---|---|---|---|
| LP | Source | Size | Source | Size | Source | Size |
| En-De | Europarl | 25000 | Flores+ | 1012 | EMEA | 500 |
| En-Hi | Samanantar | 25000 | Flores+ | 1012 | Judicial | 500 |
| En-Ta | Samanantar | 25000 | Flores+ | 1012 | Tanzil | 500 |
| En-Lt | Europarl | 25000 | Flores+ | 1012 | EMEA | 500 |

Table 2: Links to datasets used

| **Dataset** | **Link** |
|---|---|
| Europarl | `Helsinki-NLP/europarl` |
| Tanzil | `Helsinki-NLP/tanzil` |
| EMEA | `Helsinki-NLP/emea` |
| Samanantar | `ai4bharat/samanantar` |
| Flores+ | `openlanguagedata/flores_plus` |
| Judicial | `cfilt/iitb-english-hindi` |

## C.2  TEXT-TO-SQL

The paraphrases for this task are obtained using OpenAI-O3-mini. Given an NL query, SQL pair $(\mathbf{x}, \mathbf{y})$, we prompted the LLM to generate first a paraphrased SQL $\tilde{\mathbf{y}}$ that differs from $\mathbf{y}$ only in the mention of literal constants, and then generating the corresponding NL question $\tilde{\mathbf{x}}$ describing the SQL $\tilde{\mathbf{y}}$. An example paraphrased pair appears below:

> $\mathbf{x}$: Is molecule TR183 known to be a carcinogen?
> $\mathbf{y}$: SELECT T.label FROM molecule AS T WHERE T.molecule_id = 'TR183'
> .
> $\tilde{\mathbf{x}}$: Molecule KAC16 is carcinogenic. Yes or No?
> $\tilde{\mathbf{y}}$: SELECT T.label FROM molecule AS T WHERE T.molecule_id = 'KAC16'

The prompt to the LLM comprises of generic natural language instructions, followed by a description of the schema and metadata of the database queried, followed by the in-context examples, and then the current test question $\mathbf{x}_*$. Following standard practice, instead of providing the entire database metadata, we filter a subset of schema information (relevant table and column names) using an open source schema filtering tool [4] Li et al. (2023), Li et al. (2024a).

---

[4]https://github.com/RUCKBReasoning/text2sql-schema-filter

## C.3 SYNTHETIC ALIGNMENT REASONING TASK

We use the dataset as discussed by Agarwal & Sarawagi (2025) for reasoning about the alignment between two synthetic sequences. This task first defines a vocabulary $\mathcal{V}$ of symbols, each symbol $\sigma \in \mathcal{V}$ is associated with a probabilistic finite state automata (PFA) to generate sequences of length $c$ over elements over its own vocabulary. The input sequence $\mathbf{x}$ contains $m$ symbols chosen from $\mathcal{V}$, the output sequence $\mathbf{y}$ is obtained by a fixed (unknown) permutation of $\mathbf{x}$, and then sampling a length $c$ sequence from the FSA for each element in the permuted $\mathbf{x}$. More details can be found in Agarwal & Sarawagi (2025). In order to correctly generate the output for a sequence, the model needs to reason about the input-output alignment and then the PFA of each token in $\mathcal{V}$. Examples appear below:

| Prompt | Prompt string: $\mathbf{x}^1 : \mathbf{y}^1 \; \mathbf{x}^2 : \mathbf{y}^2 \; \mathbf{x}^3 : \mathbf{y}^3 \; \mathbf{x}_*$ |
|---|---|
| Example $m = 3, c = 2$ | `ACB: rijjpr CAB: jjriwp ABC: rtprjh BCA:` |
| Example $m = 2, c = 4$ | `AC: ririjhjh CA: jjhhriir AB: rttrprrp BA:` |

For this task we choose $|\mathcal{V}| = 25$ and ran under two settings of sequence lengths ($m = 4$, $c = 4$) and ($m = 6$, $c = 6$) with the alignment as a permutation. The training is done on 200 instances for 5 epochs. In the original paper, each instance sampled its own PFA to simulate an infinite mixture. In this paper, since our goal is to develop ICL-IWL mixture for a fixed task, the PFAs per symbol were fixed throughout the entire process. We sampled in-context examples at different similarity levels as follows: the Random-Context data is sampled using a vocabulary of size 25, and the Similar-Context data is created by using a vocabulary of size 12. For the Contrastive-Context data the paraphrases are generated by taking the vocabulary as the set of letter used in the target example, $\mathbf{x}_*$. Hence, the paraphrase is simple a shuffling of the letters in $\mathbf{x}_*$.

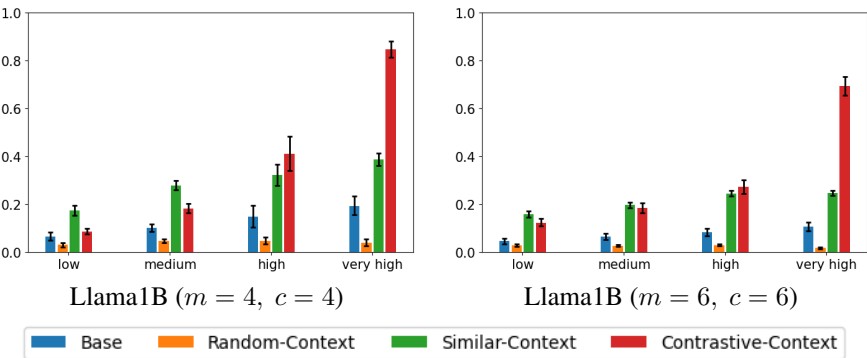

Figure 5: X-axis: Level of maximum Jaccard similarity between the sets of letters used in the target and the context examples. Y-axis: Average maximum accepted length of respective subsequence in $\mathbf{y}$ for the PFAs of each letter in $\mathbf{x}_*$. Random-Context performs worse than the baseline irrespective of the similarity level, probably because it has learnt the PFAs before learning the alignment, and thus it chooses a PFA at random for every letter in $\mathbf{x}_*$. On the other hand, the baseline model performs better because it blindly copies an ICL target example sequence corresponding to an ICL context example having some symbols in the same position as $\mathbf{x}_*$. Although Contrastive-Context performs second only to Similar-Context in the low and medium similarity regions, Contrastive-Context outperforms all the models in the high and very high similarity regions.

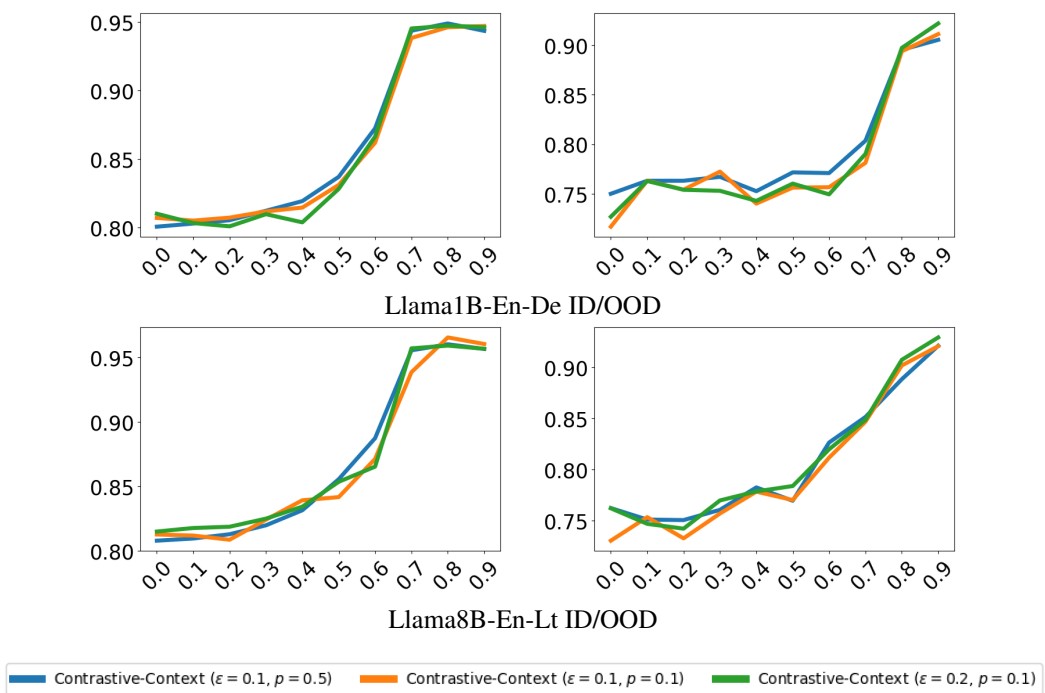

Llama1B-En-De ID/OOD

Llama8B-En-Lt ID/OOD

— Contrastive-Context ($\varepsilon = 0.1$, $p = 0.5$)   — Contrastive-Context ($\varepsilon = 0.1$, $p = 0.1$)   — Contrastive-Context ($\varepsilon = 0.2$, $p = 0.1$)

Figure 6: Contrastive-Context is robust to specific choices of $\epsilon$ and $p$ as long as sufficient contrast is maintained.

For example, in Appendix D.2, what does it mean if the performance is low in the medium similarity range but high in small and large context-target similarity, or in Sec 3.2, why does random-context has a different behavior.

## D ABLATIONS

### D.1 EXPERIMENTS WITH NON-ZERO $\epsilon$ AND $p$

We present robustness of Contrastive-Context to alternative choice of $\epsilon$ and $p$ parameters in Figure 6.

### D.2 IMPORTANCE OF TRAINING WITH DIFFERENT SIMILARITY LEVELS

Contrastive-Context includes in-context examples at three levels of similarity — randomly selected examples which are of low similarity, naturally occurring top-k most similar examples from the training pool which are typically of medium similarity levels, and highly similar examples obtained by paraphrasing the target. We establish the necessity of employing these multiple similarity levels during training by creating two ablations: first, we remove the highly similar paraphrased examples by setting $\epsilon = 0$, and second, we remove the natural similar examples by setting $p = 0$.

Figure 7 shows evaluation on test instances created with mixed similarity levels as described in Section 3.1 (Evaluation Setup). We observe that test instances with high target-context similarity suffer on models trained with $\epsilon = 0$. This is evidenced by the dip in accuracy of the **orange** line compared to the original Contrastive-Context (blue line) in the right quarter of the X-axis). Likewise, the model trained without naturally occurring Top-K pairs ($p = 0$), performs poorly on test instances where IC examples are at medium levels of similarity with the target. This is seen by the dip in accuracy of the **green** line in the middle part of the $X$ axis.

Overall, we observe the trend that it is important for IC-Train to be trained at multiple similarity levels in order to provide the best accuracy under all levels of relatedness of the test example with the context.

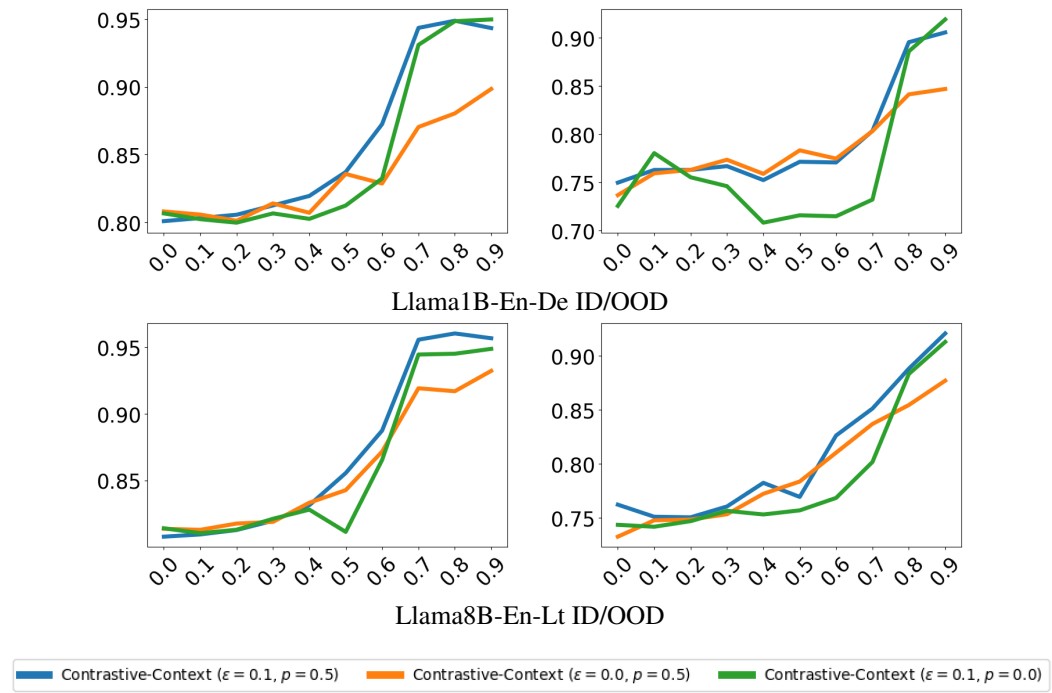

Figure 7: Ablations on Contrastive-Context to establish the importance of training with different similarity levels: $X$-axis is target-example similarity and $Y$-axis is accuracy. Removing highly similar examples obtained via paraphrasing ($\epsilon = 0$) causes test accuracy in the high similarity range to suffer. Removing natural similar examples ($p = 0$) causes accuracy in the medium range to suffer.

## D.3  INTRA-CONTEXT VS INTER-CONTEXT CONTRAST

Contrastive-Context creates contrasts in similarity levels both amongst examples within a context, and across contexts. Here we present ablations to understand the importance of contrasts within a context by comparing with a baseline that mixes Random-Context and Similar-Context to preserve only inter context contrast. We present results in Figure 8. On these tasks, the Random-Context and Similar-Context mixture (**green** bar) are as good as Contrastive-Context(**blue** bar) in the low and medium similarity regime. In the high similarity regime, because of the absence of paraphrases, this mixture looses out. When we remove the paraphrases from Contrastive-Context (**orange** bar) the two methods are equivalent on this task. However, conceptually, a model that mixes Random-Context and Similar-Context could learn to swing between ignoring the context (IWL) and blindly copying from the context based on aggregated similarity with the context. This forms a IWL+Copy mixture, which might not perform well when tested with only a subset of context examples as relevant. Contrast within a context promotes true in-context learning where similarity between the x-s determines which y-s are copied. In real-life limited data regimes, the top-K similar examples may differ in their similarity to the target, and thus Random-Context+Similar-context may behave like Contrastive-context like we observed above.

## D.4  SYNTHETIC PARAPHRASES FOR DATA AUGMENTATION VS CONTRAST

An interesting question is whether the observed accuracy boost with synthetic paraphrased examples is due to increased contrast, or vanilla data augmentation. To answer this question, we added the synthetic paraphrases for all target training examples to the training pool as augmented data. We then invoked the Similar-Context method on this augmented dataset. A comparison appears in Figure 9. We observe that with such data augmentation Similar-Context does perform well on test data in the high similarity range but shows a huge drop in accuracy in the low similarity range. This is because Similar-Context goes for the Top-K most similar examples, and all training instance contain

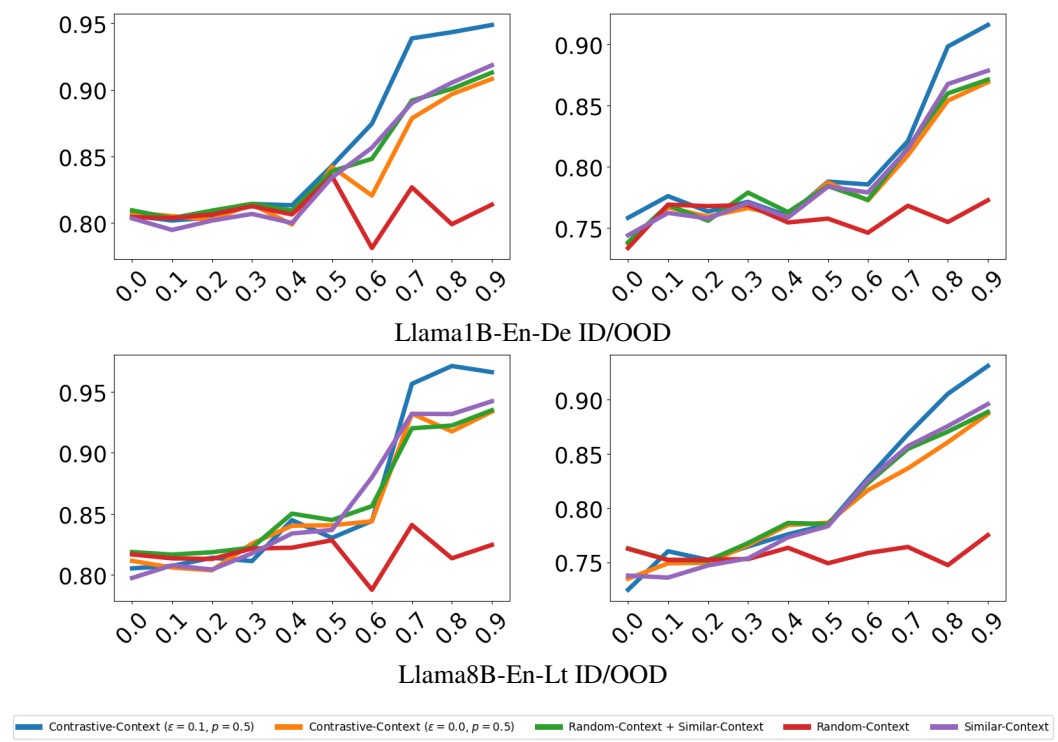

Figure 8: Comparing a Random-Context+Similar-Context mixture with Contrastive-Context that additionally also creates contrasts within a context, and other baselines Random-Context and Similar-Context.

a paraphrase in the context causing no incentive for IWL to develop. Contrastive-Context's use of paraphrases in a contrastive setting is important to develop IWL+ICL mixtures.

## D.5 EXPERIMENTS WITH VARIOUS PARAPHRASING MODELS

Although our proposed method relies on an external paraphrasing model (specifically, `gemini-2.0-flash-lite`), we demonstrate that the choice of the paraphraser has negligible impact on overall performance. To validate this, we conduct additional experiments using two alternative models `gemini-2.5-flash-lite` as an example of a strong model, and `Qwen2.5-7B-Instruct` as an open source, possibly weaker model. The results in Figure 10 shows that Contrastive-Context remains stable regardless of whether the paraphrasing model is open-source or proprietary.

## E ARCHITECTURE OF A TRANSFORMER THAT IMPLEMENTS THE MINIMAL MODEL

Here, we detail the construction of a Transformer whose final prediction matches the minimal model analyzed in Section 4.1: $\hat{\mathbf{y}} = a_* \hat{f}(\mathbf{x}_*) + \sum_{i=1}^{k} a_i y_i$. The final prediction is taken from the output embedding of the last token in the sequence (the query token).

### E.1 TOKEN REPRESENTATIONS AND PROCESSING

Let the model's internal embedding dimension be $d_{\text{model}} = 2d + 3$. For clarity, $I_n$ is the $n \times n$ identity matrix, $\mathbf{0}_{m \times n}$ is an $m \times n$ zero matrix, and $\mathbf{0}_n$ is an $n$-dimensional zero vector.

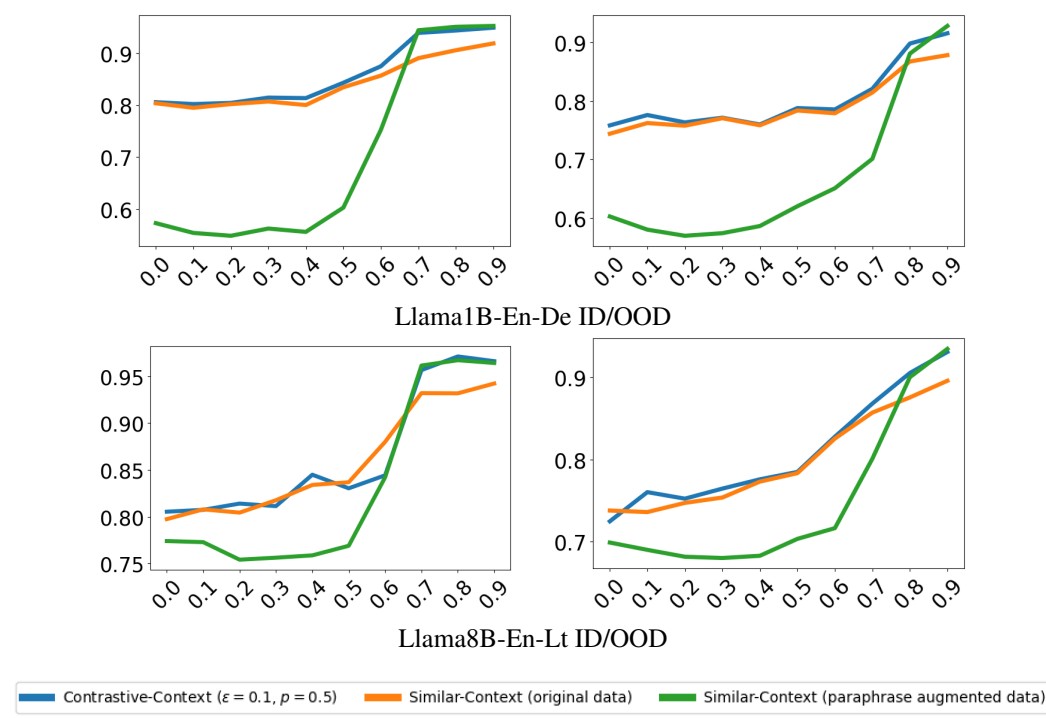

Figure 9: Comparing Similar-Context on original data and on data augmented with paraphrases. $X$-axis is similarity of test target instance with in-context examples, and $Y$ axis is accuracy. Contrastive-Context uses paraphrases selectively to create contrast, whereas Similar-Context greedily selects Top-K most similar examples. While performance of Similar-Context improves in the high similarity range, it suffers in the low similarity range because IWL does not develop with highly similar context.

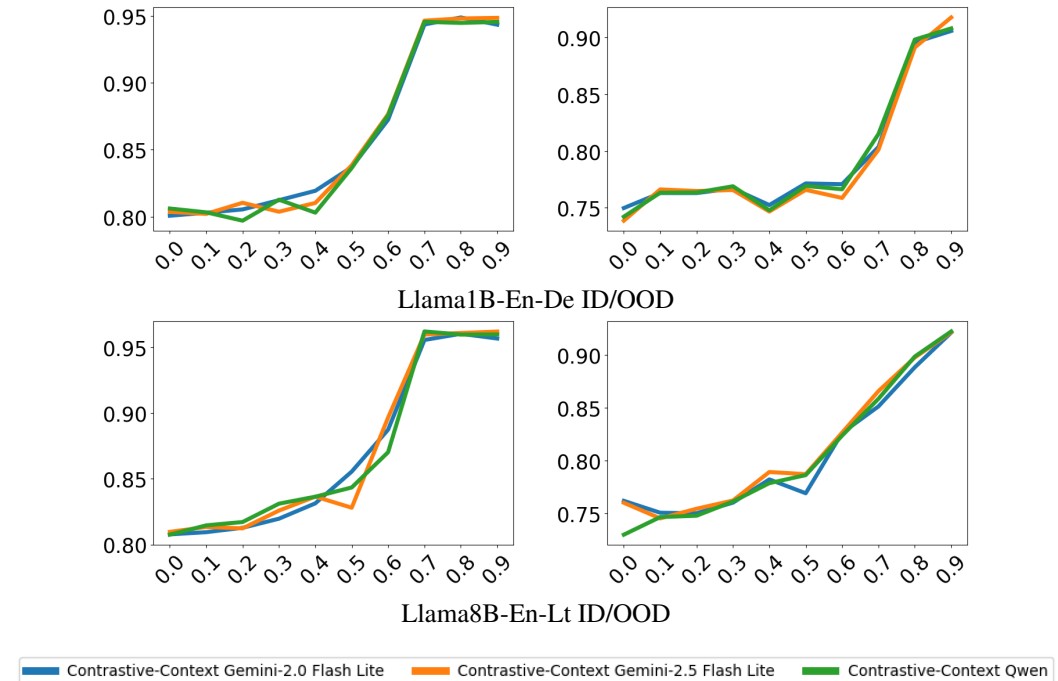

Figure 10: Comparing Contrastive-Context with highly similar examples created with various paraphrasing models. $X$-axis is target-example similarity and $Y$-axis is accuracy on the test data. Contrastive-Context is robust to the choice of the paraphrasing model.

**1. Initial Embeddings**    The input sequence consists of $k$ context tokens and one query token. Their initial embeddings are structured to segregate features.

- Context Token $i$: $\mathbf{h}_i^{\text{initial}} = [\mathbf{x}^i; 1; \mathbf{0}_d; 0; y_i] \in \mathbf{R}^{2d+3}$
- Query Token $k+1$: $\mathbf{h}_{k+1}^{\text{initial}} = [\mathbf{0}_d; 0; \mathbf{x}_*; 1; 0] \in \mathbf{R}^{2d+3}$

**2. Position-wise Feed-Forward Network (FFN)**    A position-wise FFN, acting as the in-weights estimator $\hat{f}$, is applied across all token positions. Its computation is conditional on the $(2d+2)$-th dimension of the input embedding, which serves as a gate: it is '1' for the query token and '0' for all context tokens. Consequently, the FFN's output is non-zero only for the query token, where it produces the scalar estimate $\hat{f}(\mathbf{x}_*)$. This output is then added into the final dimension of the embedding via a residual connection:

$$\hat{f}(\mathbf{x}_*) = \text{FFN}(\mathbf{h}_{k+1}^{\text{initial}})$$
$$\mathbf{h}_{k+1} = \mathbf{h}_{k+1}^{\text{initial}} + [\mathbf{0}_{2d+2}; \hat{f}(\mathbf{x}_*)]$$

The embeddings of context tokens are unchanged as their gate value is zero, effectively nullifying the FFN's contribution for those positions.

**3. Final Embeddings (Input to Attention)**    The embeddings entering the final attention layer are:

- **Context Token** $i$: $\mathbf{h}_i = [\mathbf{x}^i; 1; \mathbf{0}_d; 0; y_i]$.
- **Query Token** $k+1$: $\mathbf{h}_{k+1} = [\mathbf{0}_d; 0; \mathbf{x}_*; 1; \hat{f}(\mathbf{x}_*)]$.

E.2    PROJECTION MATRICES

We design the weight matrices to project these embeddings into query, key, and value spaces. Let $d_k = d + 1$ and $d_v = 1$.

**Query Matrix** $W_Q \in \mathbf{R}^{(d+1)\times(2d+3)}$   $W_Q$ is a block matrix that isolates the feature vector $[\mathbf{x}_*; 1]$ from the query token.

$$W_Q = \begin{bmatrix} \mathbf{0}_{(d+1)\times(d+1)} & I_{d+1} & \mathbf{0}_{(d+1)\times 1} \end{bmatrix}$$

The query vector from the final position is:

$$Q_{k+1} = \mathbf{h}_{k+1} W_Q^T = [\mathbf{x}_*; 1] \in \mathbf{R}^{d+1}.$$

Queries from context positions are not used for the final prediction.

**Key Matrix** $W_K \in \mathbf{R}^{(d+1)\times(2d+3)}$   $W_K$ extracts context features and applies the learnable parameters $\theta_1, \theta_2$.

$$W_K = \begin{bmatrix} \theta_1 I_d & \mathbf{0}_{d\times 1} & \mathbf{0}_{d\times(d+2)} \\ \mathbf{0}_{1\times d} & \theta_2 & \mathbf{0}_{1\times(d+2)} \end{bmatrix}$$

The key vectors relative to the query $Q_{k+1}$ are:

- For a context token: $K_i = \mathbf{h}_i W_K^T = [\theta_1 \mathbf{x}^i; \theta_2] \in \mathbf{R}^{d+1}$.
- For the query token: $K_{k+1} = \mathbf{h}_{k+1} W_K^T = [\mathbf{0}_d; 0] \in \mathbf{R}^{d+1}$.

**Value Matrix** $W_V \in \mathbf{R}^{1\times(2d+3)}$   $W_V$ is a simple selector for the last dimension of the embeddings, which holds the label $y_i$ for context tokens and the in-weights prediction $\hat{f}(\mathbf{x}_*)$ for the query token.

$$W_V = \begin{bmatrix} \mathbf{0}_{1\times(2d+2)} & 1 \end{bmatrix}$$

The resulting scalar value vectors are:

- For a context token: $V_i = \mathbf{h}_i W_V^T = y_i \in \mathbf{R}$.
- For the query token: $V_{k+1} = \mathbf{h}_{k+1} W_V^T = \hat{f}(\mathbf{x}_*) \in \mathbf{R}$.

### E.3   ATTENTION AND FINAL OUTPUT

The raw attention logits from the query $Q_{k+1}$ to all other tokens are:

$$\text{logit}(k+1, i) = Q_{k+1} \cdot K_i = \theta_1 \mathbf{x}_*^\top \mathbf{x}^i + \theta_2 \qquad (\text{for } i = 1, \dots, k).$$
$$\text{logit}(k+1, k+1) = Q_{k+1} \cdot K_{k+1} = 0.$$

As we use very sparse $Q$, $K$ matrices and the inputs are unit norm, the logits are not likely to blow up in magnitude. Thus we omit the standard scaling of logits by $1/\sqrt{d_k}$ for analytical clarity.

After applying the softmax, the attention weights $a_i$ (for $i = 1..k$) and $a_* \equiv a_{k+1}$ are exactly as defined in the main text. The output of the attention mechanism at the query position is a scalar, computed as the weighted sum of the scalar value vectors:

$$\hat{\mathbf{y}} = \sum_{j=1}^{k+1} a_j V_j = a_{k+1} V_{k+1} + \sum_{i=1}^{k} a_i V_i = a_* \hat{f}(\mathbf{x}_*) + \sum_{i=1}^{k} a_i y_i.$$

This completes the construction.

## F   DERIVATIONS, STATIONARITY, AND OPTIMALITY

Here we provide the detailed derivations for the stationarity and optimality of the parameter limits discussed in the main text.

### F.1   COMMON ALGEBRA

For a fixed target $\mathbf{x}_*$ and context $C$, the squared loss is

$$\ell = \left( f(\mathbf{x}_*) - \hat{\mathbf{y}} \right)^2 = \left( \frac{(1+S) f(\mathbf{x}_*) - \hat{f}(\mathbf{x}_*) - \sum_{i=1}^{k} s_i y_i}{1+S} \right)^2,$$

where $s_i = \exp\left(\theta_1 \left(\mathbf{x}_i\right)^\top \mathbf{x}_* + \theta_2\right)$ and $S = \sum_{i=1}^k s_i$. The derivative of $\ell$ with respect to a generic score $s_j$ is:

$$\frac{\partial \ell}{\partial s_j} = \frac{2\big((1+S)f(\mathbf{x}_*) - \hat{f}(\mathbf{x}_*) - \sum_i s_i y_i\big)}{(1+S)^3} \left(f(\mathbf{x}_*) - y_j\right). \tag{3}$$

Using the chain rule with $\partial s_j/\partial \theta_1 = s_j((\mathbf{x}_j)^\top \mathbf{x}_*)$ and $\partial s_j/\partial \theta_2 = s_j$, the gradients of the loss with respect to the attention parameters are:

$$\frac{\partial \ell}{\partial \theta_1} = \sum_{j=1}^k \frac{\partial \ell}{\partial s_j} s_j \left((\mathbf{x}_j)^\top \mathbf{x}_*\right), \qquad \frac{\partial \ell}{\partial \theta_2} = \sum_{j=1}^k \frac{\partial \ell}{\partial s_j} s_j.$$

The gradient of the population loss $\mathcal{L}$ is the expectation of these quantities.

### F.2 PROOF OF STATIONARITY FOR EACH REGIME

We establish that the limits described are stationary points by showing the pointwise gradients vanish. By the dominated convergence theorem (assuming bounded moments), this implies the gradient of the population loss also vanishes.

**Case 1: Random-Context.** In the limit $\theta_2 \to -\infty$, every score $s_j = \exp(\theta_1(\mathbf{x}_j)^\top \mathbf{x}_* + \theta_2) \to 0$. The gradients $\partial \ell/\partial \theta_1$ and $\partial \ell/\partial \theta_2$ are sums where each term contains a factor of $s_j$, causing the full gradient to vanish pointwise.

**Case 2: Similar-Context.** In the limit $\theta_1 + \theta_2 \to \infty$, all scores $s_j \to \infty$ at a comparable rate, so $S \to \infty$ and $s_j = O(S)$. The prefactor in Eq. (3) is $O(1/S^2)$. Each term in the gradient sum, $\frac{\partial \ell}{\partial s_j} s_j$, is therefore $O(1/S^2) \cdot S = O(1/S)$, which vanishes as $S \to \infty$.

**Case 3: One-Near-Context.** In the limit $\theta_1 \to \infty$ and $\theta_1 + \theta_2 \to \infty$, the score of the near point $s_{j^\star} \to \infty$ while all other scores $s_i \to 0$. Thus, $S \sim s_{j^\star} \to \infty$. The gradient term for $j^\star$ vanishes as $O(1/S)$, and terms for $i \neq j^\star$ vanish faster as $s_i \to 0$. The total gradient vanishes.

**Contrastive-Context.** The parameter limit is $\theta_1 \to \infty, \theta_2 \to -\infty$ such that (a) $\theta_1(1-\delta_2) + \theta_2 \to \infty$ and (b) $\theta_1 \delta_1 + \theta_2 \to -\infty$. For any draw:

- If the context is *All-random*, condition (b) implies all scores $s_i \to 0$. This matches the analysis for Case 1, and the gradient vanishes.
- If the context is *One-near*, condition (a) implies $s_{j^\star} \to \infty$ and condition (b) implies $s_{i \neq j^\star} \to 0$. This matches the analysis for Case 3, and the gradient vanishes.

Since the pointwise gradient is zero for any draw from the distribution, the limit is a stationary point of the mixed-regime loss.

### F.3 OPTIMALITY ANALYSIS

**Optimality in Random-Context.** At the stationary point, the prediction is $\hat{\mathbf{y}} \to \hat{f}(\mathbf{x}_*)$ and the population loss is $\mathcal{L}_{\text{param}} = E$. The alternative, a fully ICL prediction, would be an average of context labels where all context points are far from the target. Since $\|\mathbf{x}_i - \mathbf{x}_*\|_2^2 \geq 2(1 - \delta_1)$, the Lipschitz assumption implies an ICL-induced error of at least $2L^2(1 - \delta_1)$. Bound in (A) states $E \leq 2L^2(1 - \delta_1)$, so the parametric extreme is optimal.

**Optimality in Similar-Context.** The loss at this ICL stationary point is determined by the prediction $\hat{\mathbf{y}} \to \sum_i w_i y_i$. Since all context points are near the target ($\|\mathbf{x}_i - \mathbf{x}_*\|_2^2 \leq 2\delta_2$), the Lipschitz property implies $(f(\mathbf{x}_*) - y_i)^2 \leq 2L^2 \delta_2$. By Jensen's inequality, $\mathcal{L}_{\text{icl}} = \mathbb{E}[(f - \hat{\mathbf{y}})^2] \leq \mathbb{E}[\sum_i w_i(f - y_i)^2] \leq 2L^2 \delta_2$. The parametric loss is $\mathcal{L}_{\text{param}} = E$. Bound (A) states $E \geq 2L^2 \delta_2$. Thus, $\mathcal{L}_{\text{icl}} \leq \mathcal{L}_{\text{param}}$, making the ICL extreme optimal.

**Optimality in One-Near-Context.** At this ICL stationary point, the prediction becomes $\hat{\mathbf{y}} \to y_{j^\star}$. The loss is $\mathbb{E}[(f(\mathbf{x}_*) - y_{j^\star})^2]$. Since $\mathbf{x}_{j^\star}$ is near $\mathbf{x}_*$, $\|\mathbf{x}_{j^\star} - \mathbf{x}_*\|_2^2 \le 2\delta_2$, so by Lipschitz continuity, $\mathcal{L}_{\text{icl}} \le 2L^2\delta_2$. The parametric loss is $\mathcal{L}_{\text{param}} = E$. Bound (A) again states $E \ge 2L^2\delta_2$, so $\mathcal{L}_{\text{icl}} \le \mathcal{L}_{\text{param}}$, making this strategy optimal.

**Optimality in Contrastive-Context.** This stationary point is optimal for the mixed distribution because it dynamically selects the best strategy for each scenario. It defaults to the parametric model for All-random contexts (achieving the optimal loss $E$) and switches to ICL for One-near contexts (achieving the optimal loss $\le 2L^2\delta_2$). As it achieves the minimum possible loss for any draw from the distribution, it minimizes the expected loss over the entire distribution.

F.4   LEARNING DYNAMICS OF THE IN-WEIGHTS ESTIMATOR $\hat{f}$

We now analyze the learning signal for the in-weights estimator $\hat{f}$ during training. The prediction error can be decomposed as:

$$\hat{\mathbf{y}} - f(\mathbf{x}_*) = a_*(\hat{f}(\mathbf{x}_*) - f(\mathbf{x}_*)) + (1 - a_*)(\mathbf{y}_{\text{wavg}} - f(\mathbf{x}_*)),$$

where $\mathbf{y}_{\text{wavg}} = \sum_i (a_i/(1 - a_*))y_i$ is the weighted average of context labels. The gradient of the instantaneous loss $\ell = (\hat{\mathbf{y}} - f(\mathbf{x}_*))^2$ with respect to the output $\hat{f}(\mathbf{x}_*)$ is:

$$\frac{\partial \ell}{\partial \hat{f}(\mathbf{x}_*)} = 2(\hat{\mathbf{y}} - f(\mathbf{x}_*))\frac{\partial \hat{\mathbf{y}}}{\partial \hat{f}(\mathbf{x}_*)} = 2a_*(\hat{\mathbf{y}} - f(\mathbf{x}_*)).$$

Substituting the decomposed error, the gradient expression guiding the learning of $\hat{f}$ becomes:

$$\frac{\partial \ell}{\partial \hat{f}(\mathbf{x}_*)} = 2\big[a_*^2(\hat{f}(\mathbf{x}_*) - f(\mathbf{x}_*)) + a_*(1 - a_*)(\mathbf{y}_{\text{wavg}} - f(\mathbf{x}_*))\big].$$

This gradient reveals a tension between learning from the parametric path (first term) and being influenced by the ICL path (second term). We analyze how this dynamic plays out as the attention parameters converge in each regime.

1. **Random-Context:** In this regime, $\mathbf{y}_{\text{wavg}}$ is an average of labels from dissimilar examples, making it a poor estimator of $f(\mathbf{x}_*)$. The term $(\mathbf{y}_{\text{wavg}} - f(\mathbf{x}_*))$ is therefore large and noisy. Thus the initial updates for $\hat{f}$ may not be in the right direction. However, as training progresses and $a_*$ approaches 1, the factor $a_*(1 - a_*)$ in the second term vanishes. This silences the "polluting" influence of the noisy context. The gradient becomes dominated by the first term, $2a_*^2(\hat{f} - f) \approx 2(\hat{f} - f)$, which is precisely the gradient for a standard MSE objective. Thus, the model's learning shifts entirely to improving $\hat{f}$.

2. **Similar-Context:** Here, the context examples are highly relevant, so $\mathbf{y}_{\text{wavg}}$ is an excellent estimator of $f(\mathbf{x}_*)$ from the beginning. The term $(\mathbf{y}_{\text{wavg}} - f(\mathbf{x}_*))$ is small. So even though the initial updates for $\hat{f}$ may be good, the model can achieve low loss quickly by relying on ICL, which it does by learning to drive $a_* \to 0$. As $a_*$ decreases, both the $a_*^2$ and $a_*(1 - a_*)$ factors in the gradient shrink towards zero. The learning signal for $\hat{f}$ is rapidly suppressed, effectively halting its training as the model commits to its ICL strategy.

3. **One-Near-Context:** $\mathbf{y}_{\text{wavg}}$ may initially be imperfect. And as the model learns to increase $\theta_1$, the weights concentrate on the single near example $j^\star$, and $\mathbf{y}_{\text{wavg}}$ rapidly converges to $y_{j^\star} \approx f(\mathbf{x}_*)$. Concurrently, the model drives $a_* \to 0$. As in the previous case, this causes both terms in the gradient to vanish, suppressing updates to $\hat{f}$ once the model learns to identify and copy the relevant context.

4. **Contrastive-Context:** The training process for $\hat{f}$ becomes selective and efficient. The model is exposed to a mix of *All-random* and *One-near* contexts.
    - When presented with an *All-random* context, the dynamics from point (1) apply. The attention system learns to set $a_* \approx 1$, providing a strong, clean gradient signal to train $\hat{f}$.
    - When presented with a *One-near* context, the dynamics from point (3) apply. The attention system learns to set $a_* \approx 0$, and the gradient for $\hat{f}$ is suppressed.

This demonstrates a sophisticated division of labor: the in-weights estimator $\hat{f}$ is trained almost exclusively on the subset of data where the context is uninformative and its parametric knowledge is actually needed.

# G  ADDITIONAL EXPERIMENTS COMPARING DIFFERENT TRAINING STRATEGIES

Here we present results of more model-language pair combinations that could not be fit in Figure 1 of the main paper.

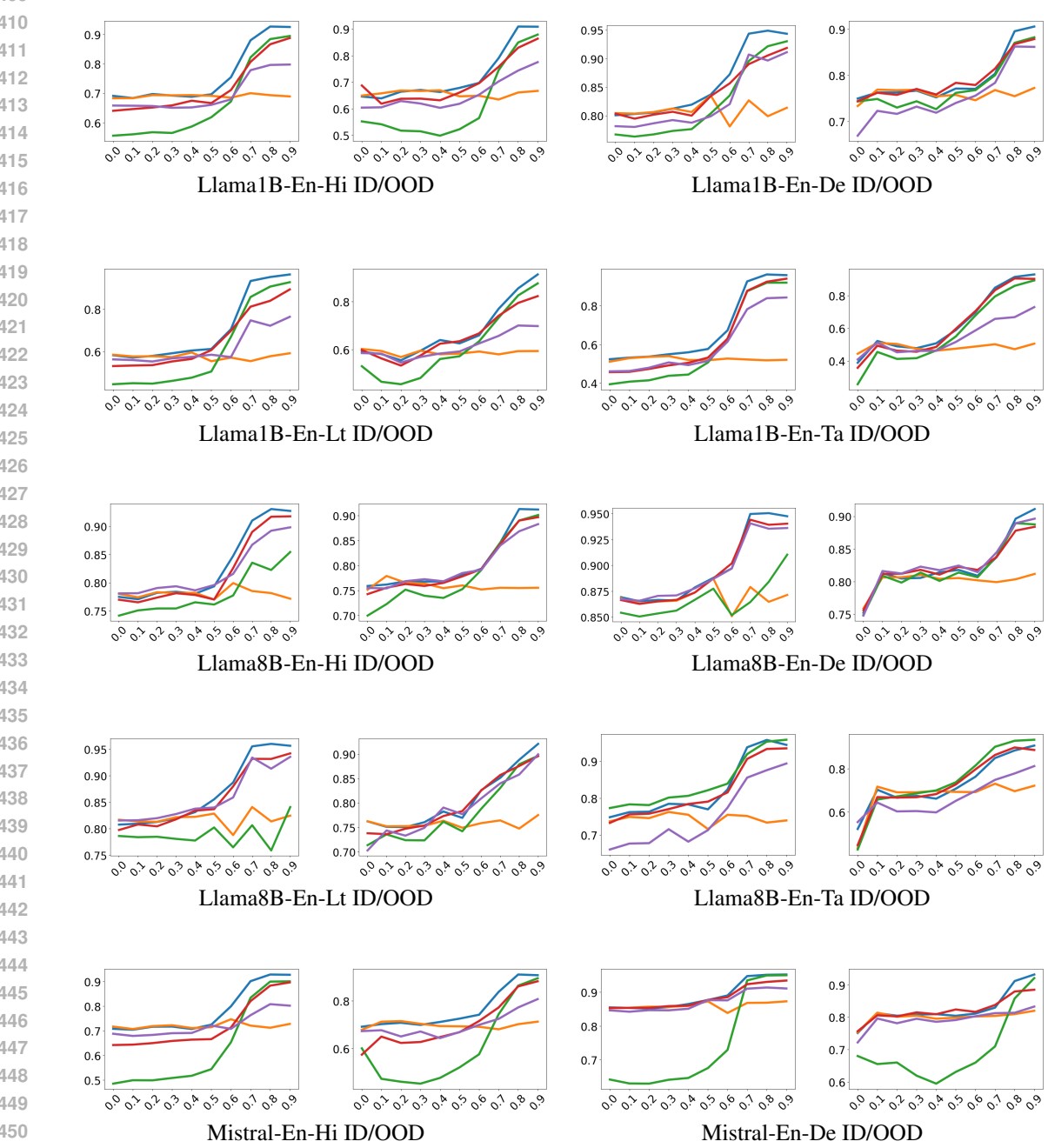

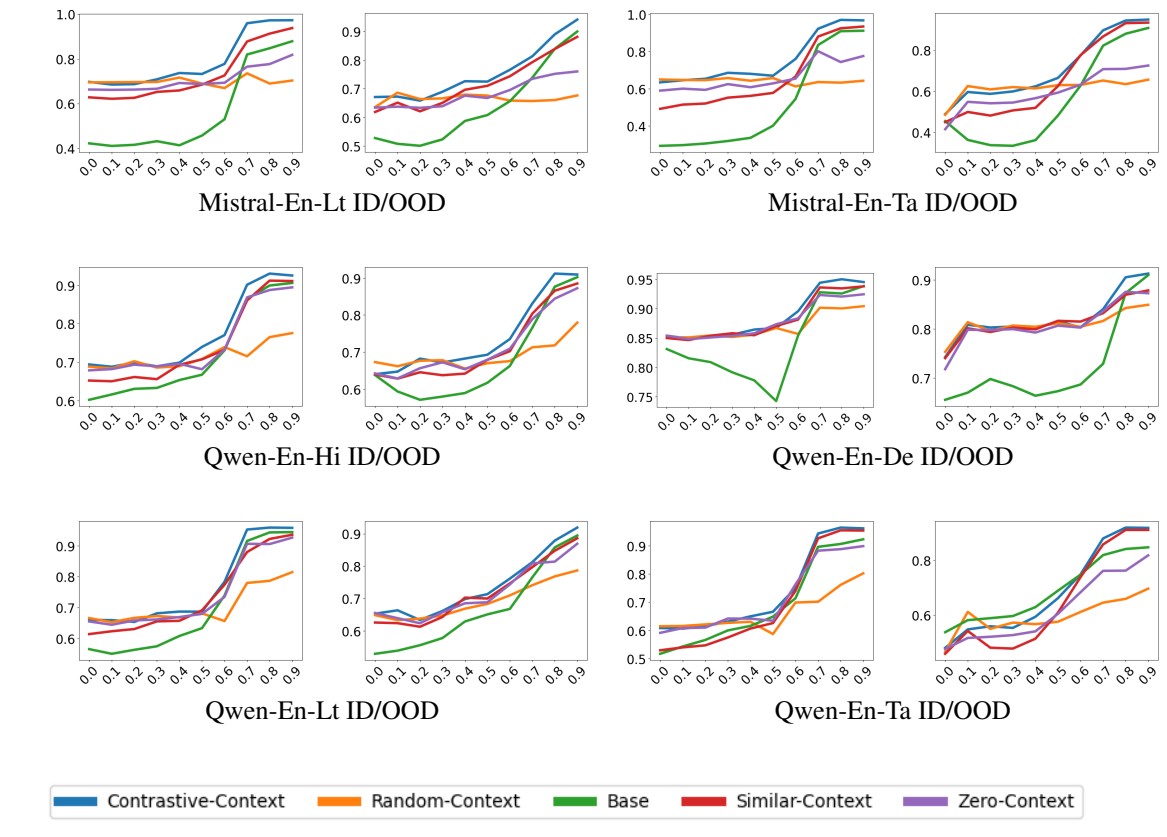

## H ADDITIONAL EXPERIMENTS ON LEARNING DYNAMICS

Here we present results of more model-language pair combinations that could not be fit in Figure 4 of the main paper.

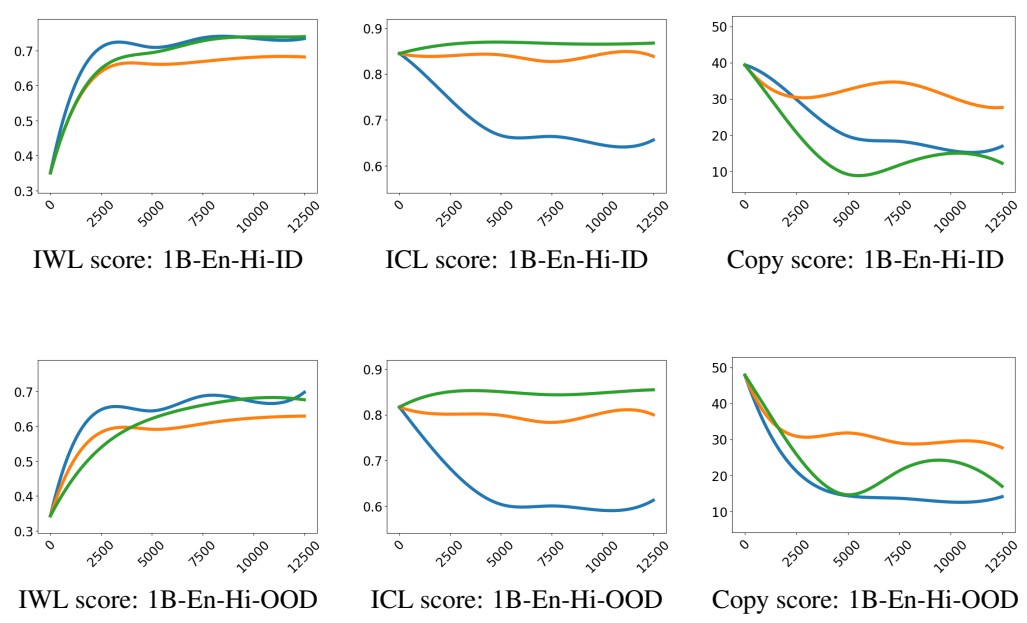

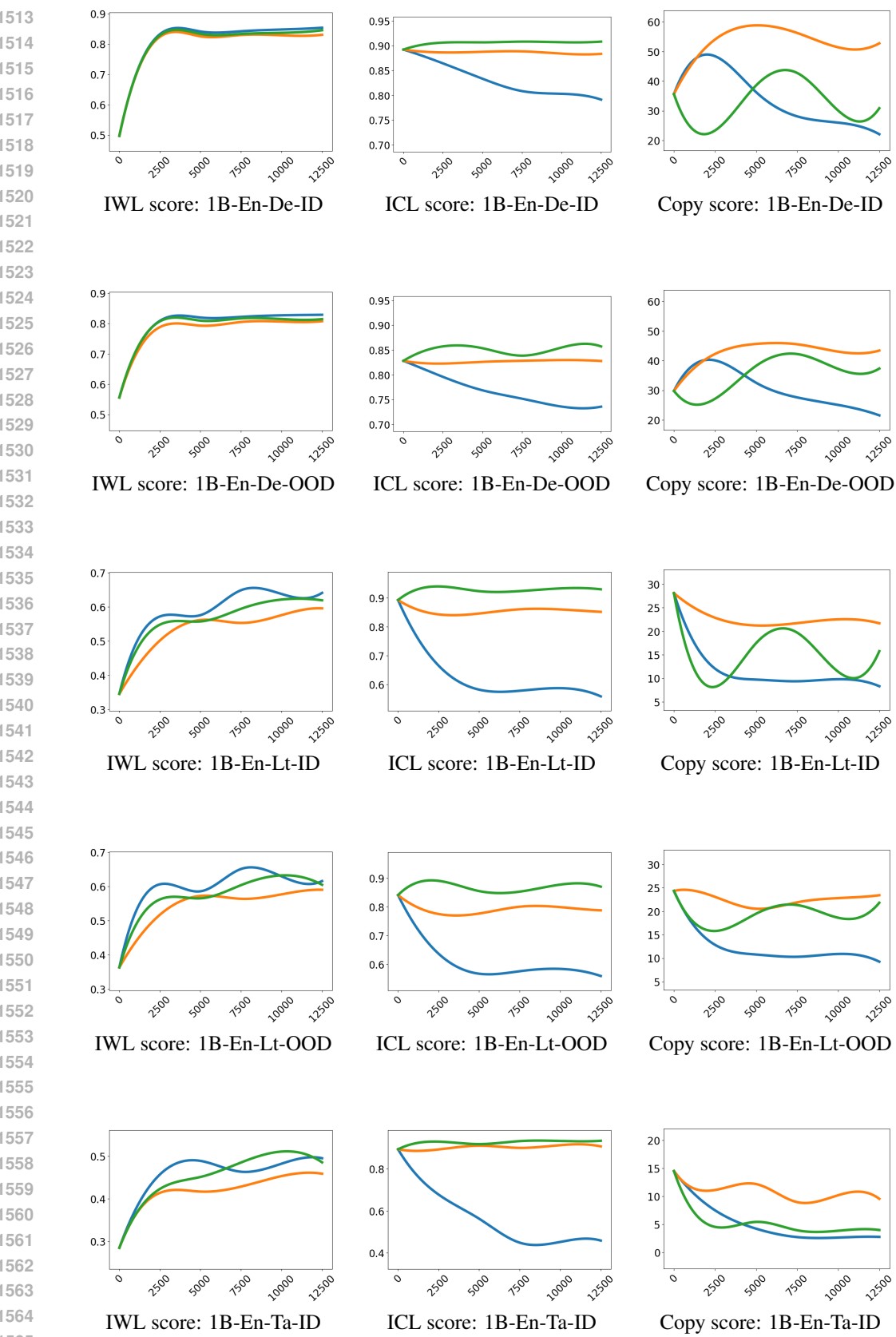

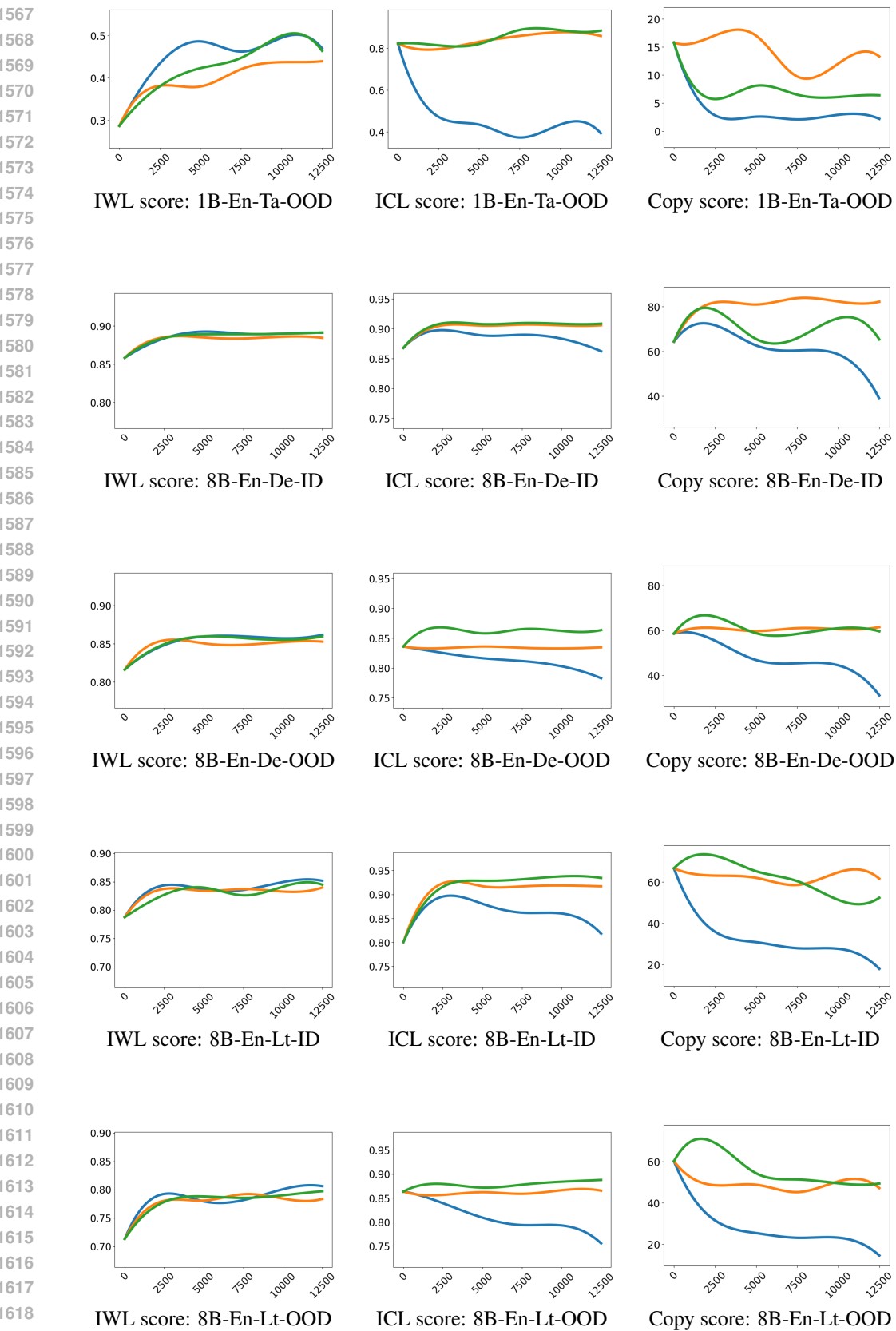

IWL score: 1B-En-Ta-OOD     ICL score: 1B-En-Ta-OOD     Copy score: 1B-En-Ta-OOD

IWL score: 8B-En-De-ID     ICL score: 8B-En-De-ID     Copy score: 8B-En-De-ID

IWL score: 8B-En-De-OOD     ICL score: 8B-En-De-OOD     Copy score: 8B-En-De-OOD

IWL score: 8B-En-Lt-ID     ICL score: 8B-En-Lt-ID     Copy score: 8B-En-Lt-ID

IWL score: 8B-En-Lt-OOD     ICL score: 8B-En-Lt-OOD     Copy score: 8B-En-Lt-OOD

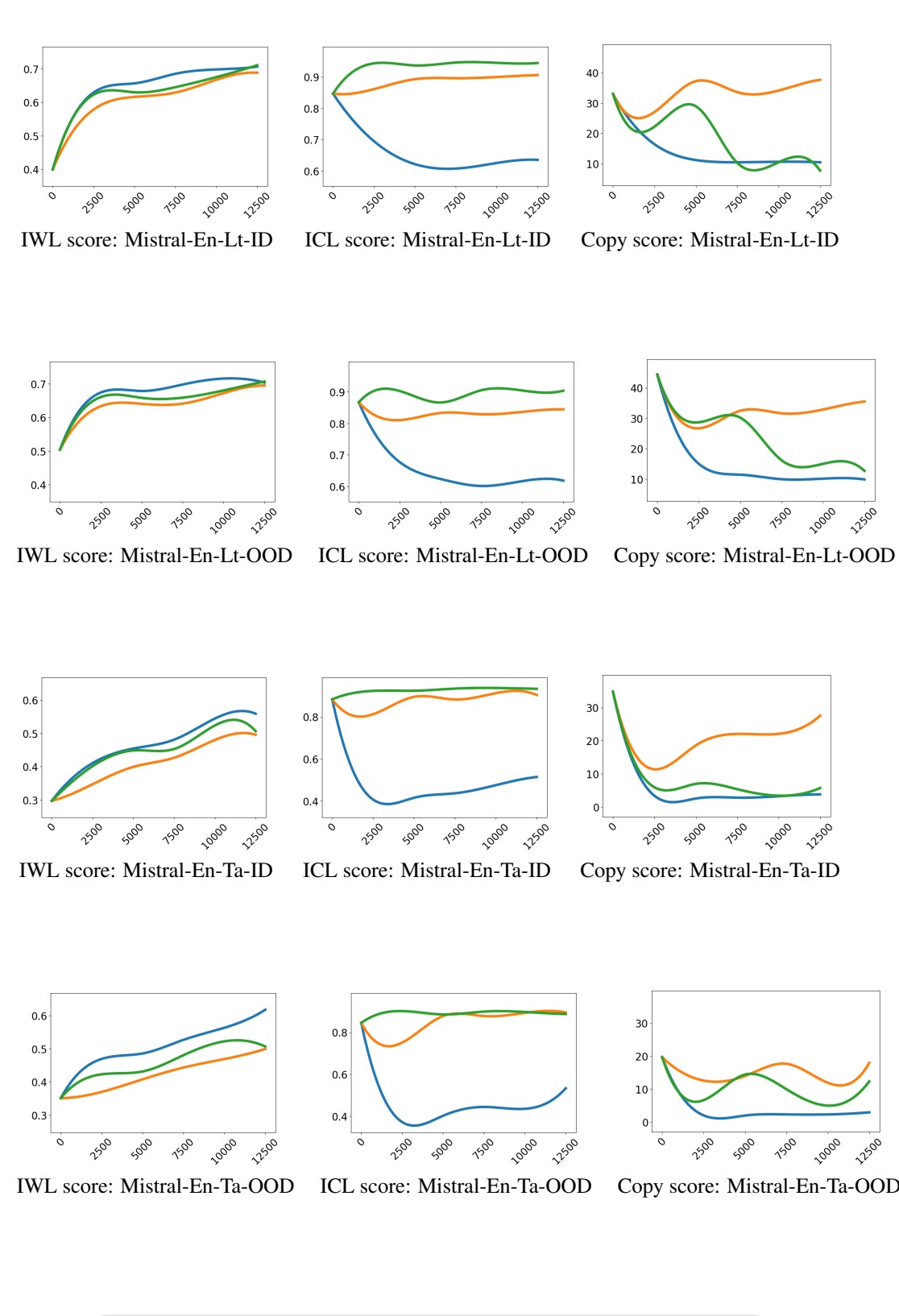

## H.1    LEARNING DYNAMICS (UNSMOOTHED)

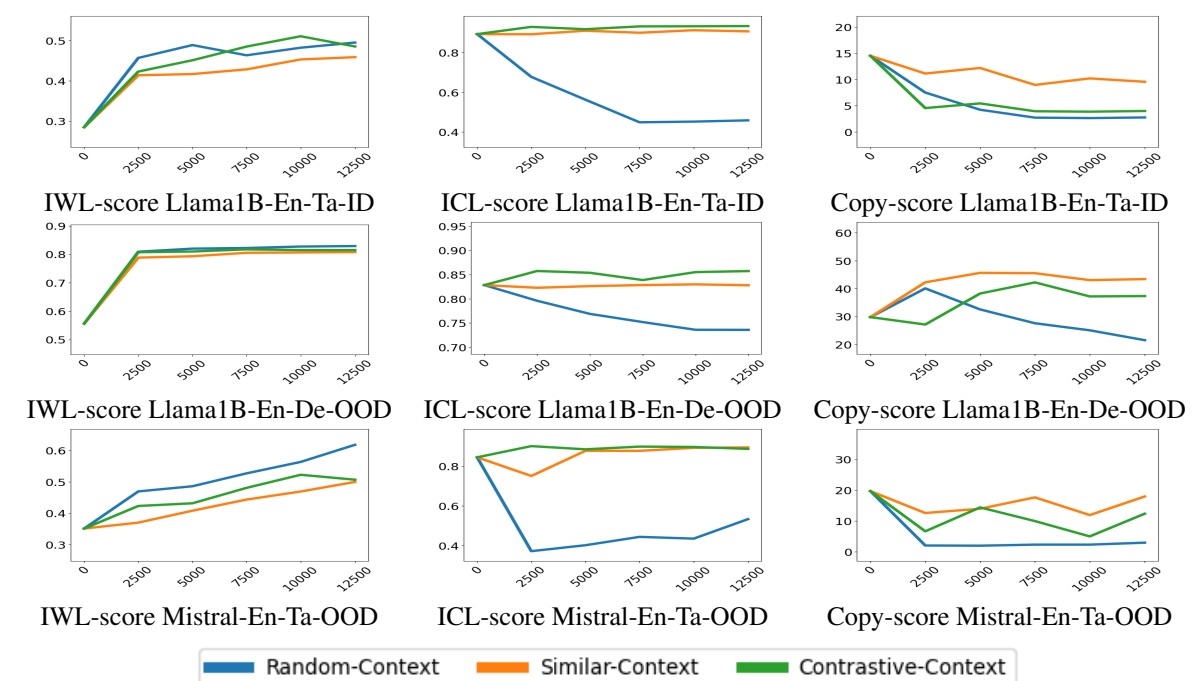

IWL-score Llama1B-En-Ta-ID     ICL-score Llama1B-En-Ta-ID     Copy-score Llama1B-En-Ta-ID

IWL-score Llama1B-En-De-OOD     ICL-score Llama1B-En-De-OOD     Copy-score Llama1B-En-De-OOD

IWL-score Mistral-En-Ta-OOD     ICL-score Mistral-En-Ta-OOD     Copy-score Mistral-En-Ta-OOD

Random-Context     Similar-Context     Contrastive-Context

