# OpenReview forum: "Context Similarity Structure Shapes the Emergence of Reliable In-Context and In-Weights Mixtures"
_ICLR.cc/2026/Conference — Submitted to ICLR 2026_

### Official Review · Reviewer_gPby · 2025-10-26

**Soundness:** 2
**Presentation:** 2
**Contribution:** 2
**Rating:** 4
**Confidence:** 3

**Summary:**

This paper investigates how structural similarities between target inputs and contextual examples influence the emergence of in-context learning (ICL) capabilities in large language models (LLMs). The authors observe that when contextual examples are overly similar, the ICL process tends to degenerate into simple label copying, disregarding the true relevance of examples. To mitigate this issue, they propose a method called Contrastive-Context, which enforces two types of contrastive structures:

(1) mixing similar and random examples within a context to encourage the development of a more robust form of ICL, and

(2) varying the degree of similarity across contexts to facilitate a balanced evolution between in-weights learning (IWL) and ICL.

Experimental results demonstrate that Contrastive-Context enhances ICL performance while maintaining IWL effectiveness, outperforming both random sampling and nearest-neighbor sampling strategies in both in-domain and out-of-domain evaluations. Furthermore, the paper provides theoretical analysis based on a two-layer Transformer model to support its empirical findings.

**Strengths:**

1. The study addresses an important and timely question—how to improve the in-context learning ability of LLMs—which is of substantial relevance to the field.
2. The authors not only introduce a novel technique, Contrastive-Context, supported by convincing experimental evidence, but also complement it with theoretical analysis and diagnostic probes that help elucidate the underlying mechanisms of their approach.

**Weaknesses:**

1. The primary limitation of this paper lies in its insufficient positioning within the broader research landscape. As a result, both its contributions and innovations remain unclear. Specifically:

   1. The claimed innovation of the paper rests on the assumption that *inter-example similarity is a critical but underexplored factor*. However, this assumption is neither adequately explained nor empirically validated. While numerous prior studies have examined how various factors influence the emergence of in-context learning (ICL), the authors do not convincingly justify why inter-example similarity should be considered more important than other factors. Moreover, the paper lacks comparative experiments that could substantiate the claimed significance of this factor.

   2. Several existing works[1,2] have proposed methods to enhance the emergence of ICL by focusing on example selection. The authors should at least include comparisons with these related approaches to clarify the novelty and relative effectiveness of their method.

        [1] Breaking through the learning plateaus of in-context learning in transformer.

        [2] Task diversity shortens the icl plateau.


   3. The paper makes several key claims that are not supported by sufficient evidence. For example:  "A natural alternative is to fine-tune in ICL mode (IC-Train), but this is challenging due to competition with in-weights learning (IWL)"[line 43-44]. and "Such adaptation should boost accuracy on inputs resembling the new examples, while retaining generalization to inputs without close neighbors." [line 37-38]



2. The experiments are conducted only on the language translations tasks. It is questionable whether the findings also apply to other area, like math or coding

**Questions:**

See weakness

---

> ### Author Response · Authors · 2025-11-25
>
> Weakness 1:
>
> A key difference in the setting of our work, and most of the previous work that study emergence of ICL is that we consider fine-tuning with a *single* real-life task.  Most of the previous studies promote task diversity as an important factor for promoting ICL, and they evaluate on synthetic datasets where sampling of diverse tasks is within their control.  In our setting, it is not possible to leverage task diversity since we are working with a single task.  In the single task setting, example similarity within and across contexts surfaces as an important overlooked factor.
>
> Weakness 2:
>
> Thanks for pointing to these papers.  We have included them in the discussion of related work in the revised version of the paper. We present here a more detailed discussion, than we had space for in the revision.
> [2] is similar to prior work that we discuss around lines 97-99 that propose task diversity as a key enabler to promote ICL.   [1] also leverages task diversity to promote ICL, but additionally also proposes to train with a mix of a fixed task (to promote IWL) along with randomly sampled variant tasks to promote ICL.   As we point out in that section, most of these works consider synthetic settings where a diverse set of tasks can be sampled as required for promoting ICL.  We consider a real-world setting of fine-tuning an LLM on a given task, where such flexibility of sampling diverse tasks to promote ICL does not apply.  We instead in this paper present an alternative novel view of focusing on example similarity within and across context to obtain the right mixture of ICL and IWL.
>
> Weakness 3:
>
> All our experiments across the 32 (model,task,test-set) combinations spanning Figures 1 and 3 and their overflow experiments in the appendix, show that fine-tuning in ICL mode entails competition between emergence of ICL and IWL.  These experiments also show that training with Contrastive-Context (1) retains generalization to inputs without close neighbors as evidenced by its top accuracy in the low similarity realms (left part of the axis in Figure 1), and (2) boosts accuracy on inputs resembling the new examples as evidenced by significantly higher accuracy than zero-shot or baseline in the high similarity range (right part of the axis in Figure 1).  We request you to please clarify if there is some other form of evidence you were considering.  We have also added results on other datasets.
>
> Weakness 4:
>
> We present experiments on additional tasks in Section 3.2 of the revision. Currently, we have added results for Text-to-SQL and observe similar trends in the relative merits of different approaches across models.  We plan to include results from more tasks in the next few days.

---

### Official Review · Reviewer_ratK · 2025-10-30

**Soundness:** 3
**Presentation:** 2
**Contribution:** 3
**Rating:** 4
**Confidence:** 4

**Summary:**

The paper studies how to fine-tune LLMs that balance in-context learning and in-weight learning, and allow switching between them based on the relevance of context examples to the target input. It provides analysis on how target-context similarity during training influences IWL and ICL balance and proposes a simple contrastive-context training strategy. They conduct an theoretical analysis with clearly defined assumptions on a regression task using a two-layer two-parameter model, to show how different contrasts in context-target influence the ICL behavior. They show that similarity structure between target and context is a key component for ICL, demonstrated on machine translation tasks over 4 open-source LLMs.

**Strengths:**

1. The problem addressed in this work is quite relevant for the community. The proposed strategy is simple and effective. Convincing empirical evidence is provided to prove its effectiveness. The proposed data strategy is tested across four LLMs at different scales over four machine translation tasks. The empirical evaluation scheme over varying grades of similarity is quite insightful. This scheme can be used for future ICL research since it captures both ICL and IWL performance as two ends on the spectrum.

2. It is a novel conceptual findings that random and similar-context pushes the model toward different extremes of ICL and IWL. Additionally, the method is theoretically grounded in a simplified two-layer model analysis, providing mechanistic understanding on how the similarity structure influences the performance.

3. The paper is well-written and text part is easy to follow.

**Weaknesses:**

1. Contrastive-context relies on synthetic paraphrases with epsilon=0.1, which is a form of data augmentation. It is unclear whether the observed improvement arises from this synthetic paraphrasing or just from the mixture of random and similar/augmented sequences. Two controlled ablations would clarify this. First, by either setting p=0 or epsilon=0 to isolate the effect. Second by varying paraphrasing strength from zero to high paraphrasing.

2. The presentation of figures could be improved. Several plots lack axis labels which makes them hard to understand, especially the bottom right plots in Figure 2. Curves in Figure 3 appear overly smoothed which reduces the transparency of the experiment. Please provide raw values as well in future.

3. The evaluation lacks robustness analysis. The method is only tested on machine translation tasks and the OOD setup represents only a mild domain shift within the same language-pair task. In Figure 3, IWL performance is higher for OOD set versus ID, implying overlap with pretrained data and undermining the OOD definition. Error bars of the different performance curves are also not provided.

**Questions:**

1. Why did the authors select only sequence to sequence task for evaluation? Seq2seq is a relatively easy real-world task compared to tasks such as QA or other reasoning tasks. It would strengthen the findings, if the benefits are also demonstrated on other difficult tasks.

2. Why do plots in Figure 3 show noticeable fluctuations over the training steps, especially for Copy-scores?

3. Recently, few works [a][b]  have shown that repetitions or augmentations of the target sample in the context leads to strong ICL and IWL performance when trained with a mixture of random and bursty sequences, similar to this paper. How does proposed contrastive-context method conceptually differ from these studies.
- [a] The emergence of sparse attention: impact of data distribution and benefits of repetition (Zucchet et al. 2025)
- [b]  What Matters for In-Context Learning: A Balancing Act of Look-up and In-Weight Learning (Bratulic et al. 2025)

---

> ### Author Response · Authors · 2025-11-25
>
> Thanks very much for your review.
>
>       Weakness 1: Ablations on Contrastive-Context
>
> We added three different kinds of ablations on Contrastive-Context over Appendix sections D.2, D.3, and D.4 in the revised version.
>
>       Weakness 2: Clarity of Figures
>
> We will improve the clarity of other figures in the second revision.
>
>       Weakness 3: In Figure 3, IWL performance is higher for OOD set versus ID
>
> In Figure 3 (now Figure 4), ID and OOD plots have been given for different language pairs. The OOD plots shown are for Llama1B En-De and Mistral En-Ta, while the ID plots are given for Llama1B En-Ta. Appendix H (Additional Experiments on Learning Dynamics) contains the custom score plots for all combinations of models and language pairs, where for the same model and language pair, the OOD scores are similar or slightly lower compared to the ID scores.
>
>       Weakness 3: OOD represents only a mild domain shift
>
> In this work the focus is on understanding and characterizing the domain shift arising out of differences in similarity structure between target and in-context examples.  This is a new kind of domain shift that has not been studied before to the best of our knowledge.  The effect of topical domain shift is partially addressed by in-context learning. The Zero-Shot numbers in Appendix G are generally lower for OOD than for ID. Please note that the axis ranges could be different.  We will fix the charts so that across ID and OOD we use the same axis.
>
>       Question 1: Experiments on other tasks
>
> We present experiments on additional tasks in Section 3.2 of the revision. Currently, we have added results for Text-to-SQL and observe similar trends in the relative merits of different approaches across models.  We plan to include results from more tasks in the next few days.
>
>       Question 2: Flactuations of copy-scores
>
> For copy-scores we used BLEU scores since lexical match seemed important for this metric.  BLEU tends to be more jumpy than COMET scores used elsewhere. This is because of the large variance of BLEU: it is usually less than 30 (for En-Ta) or 50 (for En-De) but sometimes jumps to 90–100 for near matches.
>
>       Question 3:  Related work
> Thanks for pointing these papers.  We have added a brief discussion in the related work section of the revised paper.  Here we present a longer discussion.
>
> Zucchet et al 2025  is mostly focused on sparse attention, though one section is relevant to ICL which shows that n-gram/token repetition in context speeds up the emergence of ICL, and they do not discuss much on IWL Vs ICL or the role of contrast in the context.  Also, their empirical evaluation is on synthetic tasks.
>
> Bratulic et al. 2025 has more similarities with our paper although they study only image classification tasks.  Their finding is that  using Q (query image) in the context during training boosts ICL.  This finding is similar to our conclusion on the role of paraphrases to boost ICL for sequence to sequence tasks.  The paper also discusses that higher difficulty IWL task fosters ICL, and this agrees with  what  we saw in German to English v/s English to Hindi experiments.

---

> > ### Comment · Reviewer_ratK · 2025-11-26
> >
> > Thank you for adding new ablations with different hyperparameters. In the newly added plots in Appendix D.2 and D.3, it is interesting that p=0.0, eps=0.1 performs almost as good as using most-similar examples.
> >
> > However, the work still has a few issues.
> > 1. Presentation of results is weak. The current presentation makes it difficult to assess the robustness of the findings. Since there is noticeable fluctuations in many curves, error bars should be added to the plots to judge the significance of the results.
> >
> > 2. Some trends are reported but not sufficiently discussed. For example, in Appendix D.2, what does it mean if the performance is low in the medium similarity range but high in small and large context-target similarity, or in Sec 3.2, why does random-context has a different behavior.
> >
> > 3. The paper does not position itself properly amongst similar line of work like Chan et al. 2022, Singh et al 2023 and Bratulic et al. 2025. The idea of contrastive-context appears to be a mix of burstiness (Chan et al. 2022) and copy strategy (Bratulic et al 2025), where highly-similar samples parallel burstiness and paraphrasing is similar to the copying operation. A thorough comparison should be made to disambiguate the conceptual contributions of this work.
> >
> > Chan et al. 2022 - Data Distributional Properties Drive Emergent In-Context Learning in Transformers
> > Singh et al. 2023 - The Transient Nature of Emergent In-Context Learning in Transformers
> > Bratulic et al. 2025 - What Matters for In-Context Learning: A Balancing Act of Look-up and In-Weight Learning
> >
> > Minor remark: it would be helpful for reviewers if updates are marked in a different color text.
> >
> > Overall, I like the idea and extensive evaluation proposed in this work. I think, with better positioning w.r.t. related work, improved presentation of plots and deeper interpretation of results, this work would have a higher impact. For now, I will keep my current rating.

---

### Official Review · Reviewer_ZmL2 · 2025-10-31

**Soundness:** 3
**Presentation:** 3
**Contribution:** 2
**Rating:** 4
**Confidence:** 3

**Summary:**

This paper investigates a core problem prevalent in Large Language Model (LLM) fine-tuning: how to develop and utilize In-Context Learning (ICL) while preserving the generalization capabilities obtained through In-Weights Learning (IWL). The paper proposes a new training strategy called "Contrastive-Context" which involves: (1) mixing similar and random examples within a single context; and (2) varying the grades of similarity across different contexts. Operationally, the strategy selects, with a certain probability, either a "most-similar" real example, a "highly-similar" synthetic example (i.e., a paraphrase), or a "random" example, and fills the remaining $k-1$ slots with random examples. Experiments conducted on 4 Machine Translation (MT) tasks across 4 different LLMs (1B to 8B) show that Contrastive-Context performs exceptionally well in both In-Domain (ID) and Out-of-Domain (OOD) evaluations.

**Strengths:**

1. The paper addresses a very practical and important problem: the "forgetting" or "degradation" of ICL capabilities during fine-tuning. It clearly identifies "context similarity structure" as a key factor regulating the ICL-IWL balance.
2. The claims are strongly supported through a multi-faceted approach:
  - Empirical: It is evaluated on 32 configurations (4 models x 4 tasks x 2 evaluation sets), with detailed visual analysis across the entire similarity spectrum.
  - Theoretical: A simplified two-layer Transformer model is used to mathematically analyze the parameter dynamics ($\theta_1$ and $\theta_2$) under different strategies, clearly explaining why the Random and Similar strategies fail while the Contrastive strategy succeeds.
  - Diagnostic: The three designed probes (IWL-score, ICL-score, Copy-score) serve as a novel diagnostic tool, successfully linking the behavior of black-box LLMs to the mechanisms of the theoretical model and proving that the same trade-offs and failure modes exist in large-scale models.

**Weaknesses:**

1. All empirical evaluations are concentrated on sequence-to-sequence Machine Translation (MT) tasks, lacking results on other task types (e.g., classification, reasoning, code generation).
2. Creating "highly-similar" synthetic examples relies on an external model (e.g., gemini-2.0-flash-lite) to generate high-quality paraphrases . This adds complexity and dependency to the training pipeline, and the quality of this external model becomes a confounding variable that could impact the method's effectiveness.

**Questions:**

1. How dependent is the effectiveness of the Contrastive-Context strategy on the capabilities of the external paraphrasing model (e.g., gemini-2.0-flash-lite)? If a weaker model is used, which generates lower-quality paraphrases (in terms of semantic fidelity or diversity), will the strategy's performance degrade significantly, or even fail completely?
2. How can we determine if the role of the external model (Gemini) is merely to provide the "highly-similar" samples needed for "contrast," or if it is simply providing "better data" (i.e., a form of high-quality data augmentation)?

---

> ### Author Response · Authors · 2025-11-25
>
> We thank the reviewer for the thoughtful comments.
>
>     Weakness 1: Experiments on more datasets.
>
> We present experiments on additional tasks in Section 3.2 of the revision. Currently, we have added results for Text-to-SQL and observe similar trends in the relative merits of different approaches across models.  We plan to include results from more tasks in the next few days.
>
>      Question 1: Impact of the quality of paraphrases
>
> We have ongoing experiments on paraphrases from a weaker model.  We will be sharing the results shortly.
>
>      Question 2: Role of paraphrases: contrast Vs data augmentation
>
> Interesting question.  We added the paraphrases as additional labeled data in the pool from which we run the context-selection baselines. The results are present in Appendix D.4 (Fig 8) of the revision. We observe that  Similar-Context improves for high similarity range but its performance in the low similarity range worsens because now there is even less incentive to promote IWL.

---

### Official Review · Reviewer_fzhR · 2025-10-31

**Soundness:** 2
**Presentation:** 2
**Contribution:** 2
**Rating:** 4
**Confidence:** 4

**Summary:**

This paper investigates how the similarity structure between target inputs and in-context examples affects the emergence and stability of in-context learning (ICL) and in-weights learning (IWL) in large language models (LLMs). While pre-trained LLMs can perform both ICL and IWL, standard fine-tuning often erodes ICL capabilities. The authors identify inter-example similarity as an overlooked but crucial factor in IC-Train. They introduce Contrastive-Context, a fine-tuning strategy that mixes similar and random examples within contexts. This contrast enforces models to use context information only when it is relevant and rely on in-weights knowledge otherwise. Experiments on machine translation demonstrate that Contrastive-Context consistently outperforms random and nearest-neighbor sampling.

**Strengths:**

1.	The proposed Contrastive-Context approach effectively improves the model’s ICL performance across varying levels of example similarity.
2.	The method is conceptually straightforward and easy to implement.
3.	The authors evaluate multiple base models on machine translation tasks, demonstrating the approach’s robustness and generalization across different model architectures.

**Weaknesses:**

1.	The proposed method can be viewed as a hybrid of Random-Context and Similar-Context strategies, which limits its novelty.
2.	Experiments are restricted to translation tasks, making it difficult to convincingly establish the generality of the approach across diverse task types.
3.	The presentation of the paper could be improved — for example, the structure of the introduction section could be more organized, and there are several minor typographical errors (e.g., “IC-Trainwith” in Section 3.1).

**Questions:**

1.	How does Contrastive-Context perform on a broader range of tasks beyond machine translation?
2.	How does it compare with a baseline that directly mixes Random-Context data and Similar-Context data during training, rather than creating contrast within examples in one context?
3.	It would be helpful to include ablation experiments that remove the Highly-Similar component (using only Most-Similar examples) to verify the necessity of employing multiple similarity levels during training.

---

> ### Author Response · Authors · 2025-11-25
>
> We thank the reviewer for the helpful review.
>
>      Question 1: How does Contrastive-Context perform on a broader range of tasks?
>
> We present experiments on additional tasks in Section 3.2 of the revision. Currently, we have added results for Text-to-SQL and observe similar trends in the relative merits of different approaches across models.  We plan to include results from more tasks in the next few days.
>
>       Question 2: Comparison with Random-Context+Similar-Context mixture
>
> Conceptually, a model that mixes Random-Context and Similar-Context could learn to swing between ignoring the context (IWL) and blindly copying from the context based on aggregated similarity with the context.  This forms a IWL+Copy mixture, which might not perform well when tested with only a subset of context examples as relevant.  Contrast within a context promotes true in-context learning where similarity between the x-s determines which y-s are copied.  In real-life limited data regimes, the top-K similar examples may differ in their similarity to the target, and thus Random-Context+Similar-context may indirectly behave like Contrastive-context.  We present empirical results to support these claims in Appendix D.3.
>
>       Question 3: Ablation with removing high similarity component
>
> We added results with  ϵ = 0 in Appendix D.2 of the revised version.  We observe that Contrastive-Context with non-zero ϵ performs better than with ϵ = 0 (orange line) in the high target-context similarity region.  We also study the effect of removing the medium similarity examples in Appendix D.2.  These experiments show that presenting various levels of similarity of context examples during training helps the model perform well across all levels of similarity.

---

### Author Response · Authors · 2025-12-03
**Substantial changes to address almost every concern raised by the four reviewers.**

Dear AC,
     In the revised version of the paper we have made substantial changes to address almost every concern raised by the four reviewers.

### 1. Experiments on three new tasks.
We included experiments on three new tasks. With this change we address a major cause of concern raised by all four reviewers.

### 2. New Ablation studies.
We added four ablations in Section D of the Appendix that together address
  - Questions 2 and 3 of Reviewer fzhR
  - Questions 1 and 2 and Reviewer ZmL2
  - Weakness 1 raised by Reviewer ratK

### 3. Improved related work discussion.
We have rewritten and expanded the Related Work section to respond to Reviewer ratK’s feedback and to incorporate the additional references highlighted by Reviewer gPby (Weakness 2).

### 4. Clearer plots with error-bars and result interpretation.
We redraw the plots in Figure 1 and add error bars to address the concerns of Reviewer ratK on clarity and significance.
We added greater discussion of results and provided more detailed captions to clear the misunderstanding reflected in the response of Reviewer ratK.

### 5. We clarify a few other points about writing and presentation in the response to the reviewers.

This revision, we believe, addresses all the questions and weaknesses raised by all the reviewers.
We respectfully request the AC to reconsider the rating of the paper in light of the extensive extra experiments and analysis incorporated in this revised version.

---

### Meta-Review · Area_Chair_bTTN · 2026-01-07

**Summary:**

This paper proposes a novel training framework that balances In-Context Learning (ICL) capability and In-Weights Learning (IWL) capability. To address the issue of IC-Train which erodes ICL capability, the authors propose Contrastive-Context which enforces mixing of similar and random examples within a context and varying grades of similarity. This is motivated by the finding that the similarity between target inputs and context examples affect to what extent ICL or IWL dominates the training. Extensive experiments across 4 LLMs with diverse sizes and on 4 languages validate the effectiveness of the proposed method.

Overall, this paper proposes a novel method to improve ICL performance via an interesting perspective, i.e., the similarity of context examples. The idea of context mixing is intuitive and simple to implement, while showing a clear advantage empirically. Extensive experiments across LLMs and tasks support the claim. Further, theoretical analysis using a 2-layer transformer and diagnostic probs strengthen the contribution.

Meanwhile, the reviewers agree on the following weaknesses which the authors are encouraged to address and refine their work:
- The experiments lack a larger scope, as the method is only tested on machine translation. More diverse task types should be incorporated.
- The presentation of the paper, especially the introduction should be improved.
- Lack of further experiments testing the sensitivity of quality in paraphrase examples.
- Stability issues presented in fluctuated curves and lack of error bars.
- A throrough discussion and comparison with related works, especially those sharing similar spirits is missing, limiting the novelty of the method.

**Reviewer Concerns:**

Concerns being addressed:
- Added a baseline that directly mixes Random-Context data and Similar-Context data during training.
- Ablation study by removing high-similarity contexts.
- Additional experiments validating the role of paraphrasing (pure augmentation vs contrast)

Outstanding concerns:
- The experiments lack a larger scope, as the method is only tested on machine translation. More diverse task types should be incorporated.
- The presentation of the paper, especially the introduction should be improved.
- Lack of further experiments testing the sensitivity of quality in paraphrase examples.
- Stability issues presented in fluctuated curves and lack of error bars.
- A throrough discussion and comparison with related works, especially those sharing similar spirits is missing, limiting the novelty of the method.

**Reviewer Scores:**

I don't think the reviewer scores would change fundamentally that alters the evaluation.

---

### Decision · Program_Chairs · 2026-01-26

Reject